# Differential Privacy of Cross-Attention with Provable Guarantee

**Yingyu Liang**
The University of Hong Kong
University of Wisconsin-Madison
`yingyul@hku.hk, yliang@cs.wisc.edu`

**Zhenmei Shi**
University of Wisconsin-Madison
`zhmeishi@cs.wisc.edu`

**Zhao Song**
The Simons Institute for the Theory of Computing
at the University of California, Berkeley
`magic.linuxkde@gmail.com`

**Yufa Zhou**
University of Pennsylvania
`yufazhou@seas.upenn.edu`

## Abstract

Cross-attention has become a fundamental module nowadays in many important artificial intelligence applications, e.g., retrieval-augmented generation (RAG), system prompt, guided stable diffusion, and many more. Ensuring cross-attention privacy is crucial and urgently needed because its key and value matrices may contain sensitive information about model providers and their users. In this work, we design a novel differential privacy (DP) data structure to address the privacy security of cross-attention with a theoretical guarantee. In detail, let $n$ be the input token length of system prompt/RAG data, $d$ be the feature dimension, $0 < \alpha \leq 1$ be the relative error parameter, $R$ be the maximum value of the query and key matrices, $R_w$ be the maximum value of the value matrix, and $r, s, \epsilon_s$ be parameters of polynomial kernel methods. Then, our data structure requires $\widetilde{O}(ndr^2)$ memory consumption with $\widetilde{O}(nr^2)$ initialization time complexity and $\widetilde{O}(\alpha^{-1}r^2)$ query time complexity for a single token query. In addition, our data structure can guarantee that the process of answering user query satisfies $(\epsilon, \delta)$-DP with $\widetilde{O}(n^{-1}\epsilon^{-1}\alpha^{-1/2}R^{2s}R_w r^2)$ additive error and $n^{-1}(\alpha + \epsilon_s)$ relative error between our output and the true answer. Furthermore, our result is robust to adaptive queries in which users can intentionally attack the cross-attention system. To our knowledge, this is the first work to provide DP for cross-attention and is promising to inspire more privacy algorithm design in large generative models (LGMs).

## 1 Introduction

The development of Artificial Intelligence (AI) has four stages: (1) prediction AI, e.g., ResNet [He et al., 2016] in image classification; (2) generation AI, e.g., ChatGPT [Achiam et al., 2023] in language generation; (3) autonomous agent AI, Voyager [Wang et al., 2023] autonomously plays Minecraft game [Fan et al., 2022]; (4) Artificial Generalization Intelligence (AGI). Humans have made rapid progress in generative AI, and we are excitingly heading to the third stage, the era of AI agent [Liu et al., 2023]. One prevalent application of AI agents is customized large generative models (LGMs) agents [OpenAI, 2024a], e.g., AgentGPT [GitHub, 2024a], SuperAGI [GitHub, 2024d], MetaGPT [Hong et al., 2024b,a], GPT Researcher [GitHub, 2024c] and many so on. In particular, recently, Apple Inc. introduced Apple Intelligence [Apple, 2024], signaling the integration of LGMs into physical devices. This innovation allows devices to use personal information for real-life assistance, such as entering passport numbers when booking flights or informing users of their latest

meetings. With increased AI capabilities, privacy concerns become significant, as the more personal information devices handle, the greater the potential privacy risks.

One fundamental technique used in LGMs is cross-attention [Vaswani et al., 2017], which is an essential module in retrieval-augmented generation (RAG) [Lewis et al., 2020], system prompt, guided stable diffusion, and many so on. In RAG, to be more professional, the LGMs answer user input queries by using a domain-specific database under cross-attention, which may contain specific privacy data and knowledge so that the LGMs gain additional power. For system prompts, based on cross-attention, some customized long prompts, e.g., user information or concrete rules, are concatenated before user input to follow human instructions better, which are commonly used in ChatGPT [GitHub, 2024b], Claude3 [Anthropic, 2024] and other commercial LGMs.

Consequently, protecting the privacy of domain-specific data in RAG or system prompts is crucial as they contain sensitive information about users and companies. These data and prompts are the core assets of many start-ups. However, these data and prompts can be easily recovered [Li et al., 2023b], jailbroken [Jin et al., 2024], and released [Li et al., 2023a] by user adversarial attack [Yu et al., 2024], e.g., there are 1700 tokens in ChatGPT system prompts [Patel, 2024]. These findings highlight the critical importance of robust privacy protections in LGMs, making privacy not just essential but an urgent issue that demands immediate attention.

To fundamentally preserve cross-attention privacy, we borrow the powerful tools from differential privacy (DP) [Dwork et al., 2006], which provides measurable privacy and combines with statistical machine learning seamlessly [Ponomareva et al., 2023]. Thus, in this work, we would like to ask and answer the following question,

> *How can we use differential privacy to protect the security of cross-attention in LGMs?*

Our work demonstrates that the Softmax cross-attention computation is equivalent to computing the weighted distance problem.

**Definition 1.1** (Softmax cross-attention). *Let $n$ and $m$ be the token length of the data and input query, respectively. Let $d$ be the feature dimension. Given fixed key matrix $K \in [0, R]^{n \times d}$ and fixed value matrix $V \in [-R_w, R_w]^{n \times d}$, for any input query matrix $Q \in [0, R]^{m \times d}$, the goal of the Softmax Cross-Attention Computation is to get the matrix $\mathrm{Attn}(Q, K, V) \in \mathbb{R}^{m \times d}$, which is*

$$\mathrm{Attn}(Q, K, V) := D^{-1} A V,$$

*where $A \in \mathbb{R}^{m \times n}$ satisfies $A_{i,j} := \exp(\langle Q_i, K_j \rangle / d)$ for any $i \in [m], j \in [n]$ ($Q_i$ and $K_j$ denote the $i$-th and $j$-th rows of $Q$ and $K$, respectively) and $D := \mathrm{diag}(A\mathbf{1}_n) \in \mathbb{R}^{m \times m}$ is a diagonal matrix.*

Note that $\mathsf{Softmax}(QK^\top) = D^{-1}A \in \mathbb{R}^{m \times n}$ in Definition 1.1, which is the standard function used in transformers, and usually, we call it as attention matrix. Our main theorem, presented below, provides a robust solution of cross-attention, ensuring privacy and accuracy guarantees.

**Theorem 1.2** (Main result; Informal version of Theorem 3.1). *Let $Q, K, V, \mathrm{Attn}$ be defined in Definition 1.1. Let $\alpha \in (0, 1)$ be the relative error parameter and $p_f$ be the probability of failure parameter. Let $r, s, \epsilon_s$ be the parameters of the polynomial kernel methods (Lemma D.7). Let $l = \widetilde{O}(r)$. Then, our Algorithm 1 requires $O(lndr)$ memory with $O(lnr)$ initialization time and $\widetilde{O}(\alpha^{-1}lr)$ query time, such that with probability $1 - p_f$, the output process of cross-attention satisfies $(\epsilon, \delta)$-DP and is robust to adaptive query with relative error $n^{-1}(\alpha + \epsilon_s)$ and additive error $\widetilde{O}(n^{-1}\epsilon^{-1}\alpha^{-1/2}lR^{2s}R_w r)$.*

Our main technique in Theorem 1.2 ensures that cross-attention is differentially private by using the polynomial kernel approximation method and transforming it into a weighted distance problem. We then solve the problem by summing over weighted distances (depending on the value embedding) between the query embedding and the key embedding. We build a data structure for weighted Softmax queries in Section 4.3, and we extend this data structure to handle adaptive queries using the $\epsilon_0$-net/metric entropy argument in Section 4.4. Furthermore, our error decreases as the input token length grows, diminishing both the relative and additive errors to zero.

Our contributions are as follows:

- We demonstrate that cross-attention computations are equivalent to the weighted distance problem (Section 3).

- We design a novel algorithm (Algorithm 2) that privately answers weighted Softmax queries with high probability and a concrete accuracy bound.

- Our algorithm (Algorithm 1) handles multiple cross-attention queries and is robust against adaptive query attacks (Theorem 3.1), meaning that potential attackers cannot intentionally extract information of system prompts/RAG data.

To our knowledge, this is the first work to utilize DP to protect prompts in LGMs with theoretically provable guarantees. While some have explored protecting user/system prompts with DP [Edemacu and Wu, 2024, Mai et al., 2023], they are primarily empirical and lack theoretical guarantees. Additionally, many others are working on protecting private datasets by applying DP to the fine-tuning stage of LGMs [Behnia et al., 2022, Singh et al., 2024, Liu et al., 2024b, Yu et al., 2021, Li et al., 2021, Shi et al., 2022a], which diverges from our work. The strength of DP lies in its strong, unambiguous, and concrete definition of privacy, enabling algorithm designs with provable privacy and accuracy analysis. Therefore, we believe that the theoretical aspects of DP applications in LGMs remain a highly impactful direction, and we aim to pave the way for further exploration in this area.

## 1.1 Related Work

**Differential Privacy in Data Structure and Attention.** Differential privacy (DP) is a flourishing and powerful technique that has enormous applications in the topic of private machine learning. In the era of Large Generative Models (LGMs), there are three primary approaches to ensuring privacy: (1) during the pre-training stage: to protect training data [Abadi et al., 2016, Ponomareva et al., 2023], (2) during the adaptation stage: to protect target data [Behnia et al., 2022, Singh et al., 2024, Liu et al., 2024b, Yu et al., 2021, Li et al., 2021, Shi et al., 2022a], (3) during the inference stage: to protect user/system prompts [Edemacu and Wu, 2024] and RAG data [Lewis et al., 2020]. To protect training data, DP-SGD [Abadi et al., 2016] uses DP optimizer to ensure data privacy, severing as the traditional baseline method. Recently, numerous works have aimed to improve this method by integrating DP in both the pre-training and fine-tuning stages of LGMs [Yu et al., 2021, Li et al., 2021, Golatkar et al., 2022, Behnia et al., 2022, Shi et al., 2022a, Mattern et al., 2022, Singh et al., 2024, Zheng et al., 2024, Liu et al., 2024b]. To protect user/system prompts, Edemacu and Wu [2024] provides a survey on both DP and non-DP methods. In the use of LGMs, prompting methods almost become a standard way for inference [Schulhoff et al., 2024]. Given the billions of prompt interactions daily, ensuring privacy is essential [Mai et al., 2023]. We refer readers to Appendix B for more related works.

**Roadmap.** In Section 2, we present the preliminary of differential privacy (DP) and cross-attention. In Section 3, we present the main result of our cross-attention theorem (Theorem 3.1). In Section 4, we outline the main results of our algorithms. In Section 5, we conclude our paper.

## 2 Preliminary

In this section, we give the preliminary of differential privacy (DP) and cross-attention. In Section 2.1, we describe the notations. In Section 2.2, we give definitions related to DP.

### 2.1 Notations

We use $\Pr[]$ to denote the probability. We use $\mathbb{E}[]$ to denote the expectation. We use $\mathrm{Var}[]$ to denote the variance. For two vectors $x \in \mathbb{R}^d$ and $y \in \mathbb{R}^d$, we use $\langle x, y \rangle$ to denote the inner product between $x, y$, i.e., $\langle x, y \rangle = \sum_{i=1}^{d} x_i y_i$. We use $X \subset \mathbb{R}^d$ and $|X| = n$ to mean the same thing as $X \in \mathbb{R}^{n \times d}$. Also, we denote $x_i^\top$ as the $i$-th row of $X$. We use $x_{i,j}$ to denote the $j$-th coordinate of $x_i \in \mathbb{R}^n$. We use $\mathbf{1}_n$ to denote a length-$n$ vector where all the entries are ones. We use $\|x\|_p$ to denote the $\ell_p$ norm of a vector $x \in \mathbb{R}^n$, i.e., $\|x\|_1 := \sum_{i=1}^{n} |x_i|$, $\|x\|_2 := (\sum_{i=1}^{n} x_i^2)^{1/2}$, and $\|x\|_\infty := \max_{i \in [n]} |x_i|$.

### 2.2 Differential Privacy Definitions

In this section, we give several definitions related to differential privacy (DP). We refer the reader to Dwork and Roth [2014] for more background and details on DP.

**Definition 2.1** (Neighboring dataset). *We define the two neighboring datasets as $X, X' \in \mathbb{R}^n$ such that $\|X - X'\|_1 \leq 1$, i.e., they differ on a single data point.*

**Definition 2.2** (Sensitivity). *The sensitivity of a function $f : \mathbb{R}^n \to \mathbb{R}^d$ is defined by: $\Delta := \max_{X, X' \in \mathbb{R}^n, \|X-X'\|_1=1} \|f(X) - f(X')\|_1$.*

**Definition 2.3** ($(\epsilon, \delta)$-DP). *For $\epsilon > 0, \delta \geq 0$, a randomized algorithm $\mathcal{A}$ is $(\epsilon, \delta)$-DP, if for all $\mathcal{S} \subseteq \mathrm{Range}(\mathcal{A})$ and for all $X, X'$ such that $\|X - X'\|_1 \leq 1$: $\Pr[\mathcal{A}(X) \in \mathcal{S}] \leq \exp(\epsilon) \Pr[\mathcal{A}(X') \in \mathcal{S}] + \delta$. When $\delta = 0$, the algorithm is said to have pure differential privacy.*

We mainly use the truncated Laplace mechanism, which has the following definitions.

**Definition 2.4** (Truncated Laplace distribution). *We use $\mathrm{TLap}(\Delta, \epsilon, \delta)$ to denote the Truncated Laplace distribution with pdf proportional to $\exp(-\epsilon|z|/\Delta)$ on the region $[-B, B]$, where $B = \frac{\Delta}{\epsilon} \cdot \log(1 + \frac{\exp(\epsilon)-1}{2\delta})$.*

**Fact 2.5** (Theorem 3 in Geng et al. [2020]). *Let $z$ denote a $\mathrm{TLap}(\Delta, \epsilon, \delta)$ random variable. Then we have $\mathbb{E}[z] = 0$, and $\mathrm{Var}[z] = \frac{2\Delta^2}{\epsilon^2}(1 - \delta \cdot \frac{\log^2(1+\frac{e^\epsilon-1}{2\delta})+2\log(1+\frac{e^\epsilon-1}{2\delta})}{e^\epsilon-1})$. Furthermore, if $\delta = 0$, we have $\mathrm{Var}[z] = 2\Delta^2/\epsilon^2$, meaning truncated Laplacian mechanism will be reduced to the standard Laplacian mechanism.*

**Lemma 2.6** (Laplace mechanism, [Dwork and Roth, 2014, Geng et al., 2020], see Lemma 2.2 in Andoni et al. [2023]). *Given a numeric function $f$ that takes a dataset $X$ as the input, and has sensitivity $\Delta$, the mechanism that outputs $f(X) + z$ where $z \sim \mathrm{Lap}(\Delta/\epsilon)$ is $(\epsilon, 0)$-DP. In addition, if $\epsilon, \delta \in (0, 0.5)$, $f(X) + z$, where $z \sim \mathrm{TLap}(\Delta, \epsilon, \delta)$ is $(\epsilon, \delta)$-DP. Moreover, the truncated Laplace mechanism is always accuracy up to error $B$.*

---

**Algorithm 1** Adaptive query data structure

---

1: **datastructure** DPTREESOFTMAXADAPTIVE                    ▷ Theorem 4.4
2:   **members**
3:     $\mathcal{D}_1, \ldots, \mathcal{D}_{O(r\log(dR/(\epsilon_s p_f)))}$ : DPTREESOFTMAX             ▷ Algorithm 2
4:   **end members**
5:   **procedure** INIT($X \subset [0, R]^d, n \in \mathbb{N}_+, w \in [-R_w, R_w]^n, \epsilon \in (0,1), \delta \in (0,1), \delta' \in (0,1), c \in (0, 0.1), \epsilon_s \in (0, 0.1), p_f \in (0, 0.01)$)
6:     $l \leftarrow O(r\log(dR/(\epsilon_s p_f)))$
7:     **for** $i = 1 \to l$ **do**
8:       $\mathcal{D}_i$.INIT($X, n, w, \epsilon/l, \delta/l, \delta'/l, c, \epsilon_s$)
9:     **end for**
10: **end procedure**
11: **procedure** DISTANCEQUERY($y \in [0, R]^d, \alpha \in (0, 1)$)
12:     $l \leftarrow O(r\log(dR/(\epsilon_s p_f)))$
13:     $r \leftarrow 0^l$
14:     **for** $i = 1 \to l$ **do**
15:       $r_i \leftarrow \mathcal{D}_i$.DISTANCEQUERY($y, \alpha$)
16:     **end for**
17:     **return** Median of $r$
18: **end procedure**
19: **end datastructure**

---

## 3 Main Results: Cross-Attention

In this section, we show our main result for cross-attention. Theorem 3.1 states that we can ensure the entire cross-attention module satisfies DP and is robust to adaptive queries. Our high-level idea is based on the similarity between weighted distance problem and cross-attention. For a typical weighted distance problem, we define the following: Let $w \in \mathbb{R}^n$ be the weights, $X \in \mathbb{R}^{n \times d}$ be the data matrix, where $x_i^\top$ is the $i$-th row of $X$ for $i \in [n]$, and let $y \in \mathbb{R}^d$ be the query. Suppose we need to answer $\ell_1$-distance query. We have

$$\sum_{i \in [n]} \underbrace{w_i}_{\text{weight}} \| \underbrace{y}_{\text{query}} - \underbrace{x_i}_{\text{data}} \|_1.$$

Now we introduce cross-attention. Let $Q, K, V, \mathrm{Attn}$ be defined in Definition 1.1. In a standard cross-attention process, we have access to $K$ and $V$ before inference, but not to the user input $Q$. For the cross-attention mechanism $\mathrm{Attn}$ (Definition 1.1), we aim to ensure that the matrix $AV$ satisfies DP guarantee. Let $A_{i,j} = \exp(\langle Q_i, K_j \rangle / d)$ for $i \in [m], j \in [n]$. Let $V_{j,k} \in \mathbb{R}$ be the $(j, k)$-th entry of $V$, for $j \in [n], k \in [d]$. By post-processing property (Fact C.5), to ensure that the forward output $\mathrm{Attn}(Q, K, V) = D^{-1}AV$ (Definition 1.1) satisfies DP, we only need to ensure the DP of its component $AV$[1]. The $(i, k)$-th entry of $AV$ for each $i \in [m], k \in [d]$ is computed by

$$(AV)_{i,k} = \sum_{j=1}^{n} \underbrace{V_{j,k}}_{\text{weight}} \exp(\langle \underbrace{Q_i}_{\text{query}}, \underbrace{K_j}_{\text{data}} \rangle / d), \tag{1}$$

which can be viewed as a weighted Softmax problem, where $V$ provides the weights, $Q$ is the query, and $K$ is the dataset. Thus, we choose to add noise to $K$ and $V$ based on the similarity between the weighted distance problem and cross-attention. Furthermore, we find that we can only handle one column of $V$, i.e., $V_{*,k} \in \mathbb{R}^n$, in a single data structure. Therefore, we need to initialize a total of $d$ different data structures, each with weights $V_{*,k}$ for $k \in [d]$.

Here, we present our main result below.

**Theorem 3.1** (Softmax cross-attention, informal version of Theorem J.12). *Let $Q, K, V, \mathrm{Attn}$ be defined in Definition 1.1. Let $\alpha \in (0, 1)$ be the relative error parameter and $p_f$ be the probability of failure parameter. Let $r, s, \epsilon_s$ be the parameters of the polynomial kernel methods (Lemma D.7). Let $\Gamma_{R,s} := \max_{j \in [s]} \frac{R^j}{\sqrt{j!}}$ (Definition J.3). Let $l = O(r \log(dR/(\epsilon_s p_f)))$. There is a data structure* DPTREESOFTMAXADAPTIVE *(Algorithm 1) that uses $O(lnrd)$ spaces to ensure cross-attention satisfies DP and supports the following operations:*

- *We initialize $d$ data structures using* INIT$(K, n, V_{*,k}, \epsilon \in (0, 1), \delta \in (0, 1), \delta' \in (0, 1), c \in (0, 0.1), \epsilon_s \in (0, 0.1), p_f \in (0, 0.01))$ *(Algorithm 1), for $k \in [d]$. It takes $O(lnr)$ time to initialize one data structure.*

- *At query time, for user input $Q$, we process one token at a time by passing the $i$-th row of $Q$, denoted $Q_i \in \mathbb{R}^d$, to* DISTANCEQUERY$(Q_i, \alpha \in (0, 1))$ *(Algorithm 1) for each $i \in [m]$. It takes $O(\alpha^{-1} lr \log^2 n)$ time to output an entry $z$ in $\mathrm{Attn}(Q, K, V)$ such that*

    - *the process of output $z$ satisfies $(\epsilon, \delta + \delta')$-DP,*
    - *the process of output $z$ has relative error $n^{-1}(\alpha + \epsilon_s)$,*
    - *the process of output $z$ has additive error $O(n^{-1}\epsilon^{-1}\alpha^{-1/2} l \Gamma_{R,s}^2 R_w r \sqrt{\log(l/\delta')} \cdot \log^{3/2} n)$,*
    - *it holds with probability $1 - p_f$ (where $p_f$ is used in $l$),*
    - *it is robust to adaptive query.*

In Theorem 3.1, we use our DPTREESOFTMAXADAPTIVE (Algorithm 1) and guarantee that, for each query token of cross-attention, the output process satisfies $(\epsilon, \delta)$-DP with $n^{-1}(\alpha + \epsilon_s)$ relative error and $O(n^{-1}\epsilon^{-1}\alpha^{-1/2} l \Gamma_{R,s}^2 R_w r \sqrt{\log(l/\delta')} \cdot \log^{3/2} n)$ additive error, and $O(\alpha^{-1} lr \log^2 n)$ running time under adaptive query. More specifically, the algorithm creates $d$ DPTREESOFTMAX-ADAPTIVE data structures, each requiring $O(lnr)$ memory consumption and $O(lnr)$ initialization time. Notably, our error is inversely proportional to $n$, meaning that as the input token length increases, both the relative and approximate errors approach zero. This is achieved by the normalizing matrix $D$ (Definition 1.1). We refer the reader to Section J for proof details.

Thus, our algorithm theoretically protects system prompts/RAG data in cross-attention as discussed in Section 1. In Section 4, we provide a detailed technical overview, and in Section A, we will present self-attention and DP-related discussion.

# 4 Key Data Structure: DPTree

This section provides our key data structures: DPTREE (Algorithm 3), DPTREEDISTANCE (Algorithm 5 and 6), DPTREEHIGHDIM (Algorithm 7), DPTREESOFTMAX (Algorithm 2), and DP-TREESOFTMAXADAPTIVE (Algorithm 1).

---

[1]$D$ is only a normalization factor and does not have sensitive information.

In Section 4.1, we provide our high-level proof insights. In Section 4.2, we give our basic building block algorithms DPTREE, DPTREEDISTANCE and DPTREEHIGHDIM. In Section 4.3, we present our DPTREESOFTMAX algorithm that solves the weighted Softmax problem. In Section 4.4, we present our DPTREESOFTMAXADAPTIVE algorithm that enables DPTREESOFTMAX to handle adaptive query problem.

## 4.1 Technique Overview

Notice that Eq. (1) is not a typical distance measure like $\ell_1$ or $\ell_2$, but by using polynomial kernel method techniques, we transform it into a distance measure. Alman and Song [2023] states that the exponential inner product can be approximated by polynomial kernel function $P(\cdot) : \mathbb{R}^d \to \mathbb{R}^r$, i.e., $P(x)^\top P(y) \approx \exp(x^\top y/d)$ for two vector $x, y \in \mathbb{R}^d$, with a relative error. Then, by the Law of Cosines, we transform the inner product of polynomial kernel functions into a distance measure, i.e.,

$$2P(x)^\top P(y) = -\|P(x) - P(y)\|_2^2 + \|P(x)\|_2^2 + \|P(y)\|_2^2. \tag{2}$$

After transforming Eq. (1) into a distance measure, we design the DPTREE series data structures to provide cross-attention DP guarantee.

In summary, we first design the data structure DPTREE (Algorithm 3) that builds a binary segment tree with truncated Laplace noise added in the leaf nodes to ensure DP guarantee. Then, based on this data structure, we design DPTREEDISTANCE (Algorithm 5 and 6) to answer one dimensional weighted distance queries $\sum_{i=1}^{n} w_i \cdot |y - x_i|$, which utilizes DPTREE to store and return noised weights $w_i$ multiplied with the approximated distances between the query $y$ and data $x_i$. We further decompose high dimensional $\ell_p^p$-distance problem into one dimensional $\ell_1$-distance problems using

$$\sum_{i=1}^{n} w_i \cdot \|y - x_i\|_p^p = \sum_{k=1}^{d} \sum_{i=1}^{n} w_i \cdot |y_k - x_{i,k}|^p. \tag{3}$$

Based on this decomposition, we design DPTREEHIGHDIM (Algorithm 7) which is capable of answering high dimension queries. Then, using Eq. (2) and DPTREEHIGHDIM, we design DPTREESOFTMAX (Algorithm 2) to answer Softmax queries. By building multiple copies of this data structure, we boost the success probability such that it can answer any query (including adaptive query) with an additive error, establishing the final data structure DPTREESOFTMAXADAPTIVE (Algorithm 1). See Section D for a more detailed outline of algorithms and proof techniques.

## 4.2 DPTree, DPTreeDistance, and DPTreeHighDim

We design a basic data structure DPTREE (Algorithm 3) that answers summation queries by a summation segment tree with truncated Laplace noise (Definition 2.4). The algorithm first builds a binary summation tree in an array and then adds truncated Laplace noises to each node. In query time, we first trace from bottom nodes to find their lowest common ancestor, then report the summation by using at most $2 \log n$ nodes on the path (Algorithm 3). Based on the parallel composition rule of DP (Fact C.7), we find that if we have multiple disjoint interval queries, the error of the weighted sum of the intervals can be bounded independently of the number of queries (Lemma E.8). See more details in Section E.

We then design DPTREEDISTANCE, a one-dimensional weighted $\ell_1$ distance data structure detailed in Algorithm 5 and 6. Initialization involves rounding each data point to the nearest multiple of a small interval and aggregating their weights into an array (illustrated in Figure 1), which is then input into our DPTREE. At query time, we retrieve aggregated weights within small intervals and multiply them by their distances to the query point. We introduce a relative error parameter $\alpha$ to reduce the number of iterations to $O(\log(n)/\alpha)$, improving efficiency. Guided by Eq.(3), we design DPTREEHIGHDIM (Algorithm 7), which extends DPTREEDISTANCE to higher dimension by constructing independent data structures for each coordinate. See details in Section G and H.

## 4.3 Softmax Activation

In this section, we present DPTREESOFTMAX (Algorithm 2) that answers the weighted Softmax query (Definition 4.1) and is further used to design DP cross-attention. First, we introduce the definition of weighted Softmax query, an abstraction for the problem described in Eq. (1).

---

**Algorithm 2** Softmax query

---

1: **datastrucutre** DPTREESOFTMAX            ▷ Theorem 4.2
2: **members**
3:      $\mathcal{D}_1, \ldots, \mathcal{D}_r$ : DPTREEDISTANCE            ▷ Algorithm 5, Theorem J.7
4:      $P : [0, \Gamma_{R,s}]^{n \times r}$            ▷ Definition J.3 for $\Gamma_{R,s}$, Eq. (9) for $s$, Eq. (10) for $r$
5:      $w : [-R_w, R_w]^n$
6:      $P_{wx}, \; s_w, \; \epsilon_s : \mathbb{R}$
7: **end members**
8: **procedure** INIT($X \subset [0, R]^d$, $n \in \mathbb{N}_+$, $w \in [-R_w, R_w]^n$, $\epsilon \in (0,1)$, $\delta \in (0,1)$, $\delta' \in (0,1)$,
     $c \in (0, 0.1), \epsilon_s \in (0, 0.1)$)            ▷ Lemma D.7
9:      $\epsilon_s, \; w, \; P, \; P_{wx}, \; s_w \leftarrow \epsilon_s, \; w, \; 0^{n \times r}, \; 0, \; 0$
10:      **for** $j = 1 \rightarrow n$ **do**
11:          Compute $P(x_j)$            ▷ Polynomial kernel function $P(\cdot)$, Lemma J.5
12:          Compute $w_j \|P(x_j)\|_2^2$
13:          $P_{wx} \leftarrow P_{wx} + w_j \|P(x_j)\|_2^2$
14:          $s_w \leftarrow s_w + w_j$
15:          $P_{j,:} \leftarrow P(x_j)$
16:      **end for**
17:      **for** $i = 1 \rightarrow r$ **do**
18:          $\mathcal{D}_i$.INIT($P_{:,i}, n, w, c\epsilon/\sqrt{r \log(1/\delta')}, \delta/r$)            ▷ Algorithm 5
19:      **end for**
20: **end procedure**
21: **procedure** DISTANCEQUERY($y \in [0, R]^d$, $\alpha \in (0, 1)$)            ▷ Lemma D.7
22:      Value $\leftarrow 0$
23:      Compute $P(y)$
24:      Compute $\|P(y)\|_2^2$
25:      **for** $i = 1 \rightarrow r$ **do**
26:          Value $\leftarrow$ Value $+ \mathcal{D}_i$.DISTANCEQUERY($P(y)_i, \alpha$)            ▷ Algorithm 6
27:      **end for**
28:      Value $\leftarrow 0.5 \cdot (P_{wx} + s_w \|P(y)\|_2^2 \; - \text{Value})$
29:      **return** Value
30: **end procedure**
31: **end datastrucutre**

---

**Definition 4.1** (Weighted Softmax query (without normalization))**.** *For the dataset $X \in [0, R]^{n \times d}$ where $x_i^\top$ is the i-th row of $X$ and query $y \in [0, R]^d$, we define the weighted exponential inner product/Softmax query to be:*

$$\sum_{i \in [n]} w_i \exp(\langle x_i, y \rangle / d) = w^\top \exp(Xy/d).$$

Building on Definition 4.1, we develop a novel algorithm to answer differentially private weighted Softmax queries using the polynomial kernel method from Alman and Song [2023]. Specifically, in Eq.(2), there is a term that computes the weighted $\ell_2^2$ distance, which we calculate using DP-TREEHIGHDIM. We then compute the exact term for the weighted $\ell_2^2$ norms of the approximation kernel. By summing these terms with a controlled error, we extend DPTREEHIGHDIM to answer the Softmax query efficiently. More details can be found in Section J.

**Theorem 4.2** (Softmax query, informal version of Theorem J.8)**.** *Let $R \geq 1$. Let $r \leq \binom{2s+2d}{2s}$ and $s = O(\max\{\frac{\log(1/\epsilon_s)}{\log(\log(1/\epsilon_s)/R)}, R^2\})$. Let $\Gamma_{R,s} := \max_{j \in [s]} \frac{R^j}{\sqrt{j!}}$ (Definition J.3). Let the accuracy parameter be $\epsilon_s \in (0, 0.1)$. Our data structure DPTREESOFTMAX (Algorithm 2) uses $O(nr)$ spaces to solve Softmax query problem for dataset $X \subset [0, R]^d$ and support following operations:*

- INIT($X \subset [0, R]^d, n \in \mathbb{N}_+, w \in [-R_w, R_w]^n, \epsilon \in (0,1), \delta \in (0,1), \delta' \in (0,1), c \in (0, 0.1), \epsilon_s \in (0, 0.1)$). *(Algorithm 2) It takes $O(nr)$ time to initialize the data structure.*

- DISTANCEQUERY($y \in [0, R]^d, \alpha \in (0, 1)$). *(Algorithm 2) It takes $O(\alpha^{-1} r \log^2 n)$ time to output a number $z$ such that*

- *the process of output $z$ satisfies $(\epsilon, \delta + \delta')$-DP private, which computes $w^\top \exp(Xy/d)$,*
- *the error bound satisfies $|z - w^\top \exp(Xy/d)| \leq (\alpha + \epsilon_s) \cdot w^\top \exp(Xy/d) + O(\epsilon^{-1}\alpha^{-1/2}\Gamma_{R,s}^2 R_w r \sqrt{\log(1/\delta')} \cdot \log^{3/2} n)$,*
- *it holds with probability at least $0.99$.*

**Remark 4.3.** *In Theorem 4.2, the parameter $\epsilon_s$ is the accuracy parameter for polynomial kernel approximation described in Section D.5. Besides, note that the error bound in Theorem 4.2 does not depend on $\delta$ but depends on $\delta'$. The role of $\delta$ is to control a hidden constant term in the big $O$ notation, i.e., increasing $\delta$ reduces the error by a small constant (Fact 2.5). In practice, we set $\delta$ as a small positive constant close to $0$. Please refer to the Lemma E.6 for more details.*

## 4.4 Adaptive Query Data Structure

We adapt our DPTREESOFTMAX to DPTREESOFTMAXADAPTIVE (Algorithm 1) to solve the adaptive query problem. By proving it can handle any query within the query space with a certain error, we ensure it effectively processes adaptive queries. We first boost the constant probability to high probability using the Chernoff bound (Lemma C.2). Employing an $\epsilon_0$-net argument and the union bound, we bound all query points within the net. Finally, we use the Lipschitz property of the weighted Softmax distance function with an additive error to bound all points in the query space. The corresponding proofs can be found in Section I and Section J.

**Theorem 4.4** (Adaptive query Softmax data structure, informal version of Theorem J.11)**.** *Let $R \geq 1$. Let $r \leq \binom{2s+2d}{2s}$ and $s = O(\max\{\frac{\log(1/\epsilon_s)}{\log(\log(1/\epsilon_s)/R)}, R^2\})$. Let $\Gamma_{R,s} := \max_{j \in [s]} \frac{R^j}{\sqrt{j!}}$ (Definition J.3). Let the accuracy parameter be $\epsilon_s \in (0, 0.1)$. Let $X \in [0, R]^{n \times d}$ be the dataset, $w \in [-R_w, R_w]^n$ be weights, $y \in [0, R]^d$ be the query, $\alpha \in (0, 1)$ be the relative error parameter and $p_f$ be the failure probability parameter. Let $l = O(r \log(dR/(\epsilon_s p_f)))$. There is a data structure DPTREESOFTMAXADAPTIVE (Algorithm 1) that uses $O(lnr)$ spaces to solve the weighted Softmax query problem for the dataset $X \subset [0, R]^d$ and supports the following operations:*

- INIT($X \subset [0, R]^d, n \in \mathbb{N}_+, w \in [-R_w, R_w]^n, \epsilon \in (0, 1), \delta \in (0, 1), \delta' \in (0, 1), c \in (0, 0.1), \epsilon_s \in (0, 0.1), p_f \in (0, 0.01)$)*. It takes $O(lnr)$ time to initialize the data structure.*

- DISTANCEQUERY($y \in [0, R]^d, \alpha \in (0, 1)$)*. It takes $O(\alpha^{-1}lr \log^2 n)$ time to output a number $z$ such that*

  - *the process of output $z$ satisfies $(\epsilon, \delta + \delta')$-DP private, which computes $w^\top \exp(Xy/d)$,*
  - *the error bound satisfies $|z - w^\top \exp(Xy/d)| \leq (\alpha + \epsilon_s) \cdot w^\top \exp(Xy/d) + O(\epsilon^{-1}\alpha^{-1/2}l\Gamma_{R,s}^2 R_w r \sqrt{\log(l/\delta')} \cdot \log^{3/2} n)$,*
  - *it holds with probability at least $1 - p_f$ (where $p_f$ is used in $l$),*
  - *it is robust to adaptive query.*

**Remark 4.5.** *We describe the parallelization of our algorithms. In the second for loop of INIT and the for loop of DISTANCEQUERY in Algorithm 2, the $r$ DPTREEDISTANCE data structures instantiated for each coordinate are independent of each other. In addition, the for loops in Algorithm 1 are also parallelizable since the $l = O(r \log(dR/(\epsilon_s p_f)))$ copies are independent. After parallelization, we have the final time complexity of INIT to be $O(nr)$ and DISTANCEQUERY to be $O(\alpha^{-1} \log^2 n)$ in Algorithm 1 with $O(lr)$ GPU process.*

## 5 Conclusion and Discussion

To our knowledge, we are the first work to provide differential privacy for cross-attention. This paper presents the DPTREE data structures, which provide a differential privacy guarantee for the cross-attention module in large generative models. This is achieved by transforming the cross-attention mechanism into a weighted distance problem. Furthermore, our algorithm is robust to adaptive queries, allowing users to interact with the model arbitrarily without extracting sensitive information from the system prompts or RAG data. Our results may inspire more privacy algorithm design in large generative models. Further discussion, including extension to self-attention and DP related design issue, is deferred to Section A.

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

# Appendix

## Contents

**Roadmap.** The appendix is organized as follows. In Section A, we discuss DP-related topics and potential extensions. In Section B, we provide more related works. In Section C, we give the preliminary of our paper. In Section D, we offer an outline of our proof techniques. In Section E, we give the analysis of the data structure DPTREE that can solve summation problem with DP and accuracy guarantee. In Section F, we show how to solve weighted distance problem. In Section G, we give our DPTREEDISTANCE data structure that can solve one dimensional $\ell_1$ distance problem with DP and accuracy guarantee. In Section H, we present the analysis of our DPTREEHIGHDIM (Algorithm 7) data structure, which can address the high-dimensional $\ell_1$ distance problem while ensuring differential privacy and accuracy guarantees. In Section I, we show how we can handle adaptive query. In Section J, we show how to extend our algorithm to Softmax activation and give the analysis of DPTREESOFTMAX (Algorithm 2) and DPTREESOFTMAXADAPTIVE (Algorithm 1).

## A  Discussion

**How do we extend to self-attention?** As self-attention is a more fundamental module in LGMs, we would like to extend our data structure to this setting. However, the challenge we faced was the dynamic update in tree nodes for each query for self-attention, which our current analysis does not support. How we can solve this challenge is crucial, and we leave it as our future direction.

**Why not add noise to some other places?** Where and how to add DP noises is an important problem to ask during the DP algorithm design. In this paper, we consider the problem of $\sum_{i=1}^{n} w_i \exp(\langle x_i, y \rangle / d)$ where $y, x_i \in [0, R]^d$ and $w \in [-R_w, R_w]^n$ (Definition 4.1). Notice that the only place where we add noises is in the most basic building block data structure DPTREE (Algorithm 3). From Lemma D.3 and the way we initialize DPTREE in Algorithm 5, we see that the sensitivity $\Delta$ of this problem is $2R_w$.

A simple method for adding noise involves adding $n$ noises to a length $n$ array, with each item $w_i \exp(\langle x_i, y \rangle / d)$ for $i \in [n]$. However, this approach increases the error by a factor of $n$ by basic composition (Fact C.6) and also makes the model dependent on the number of queries. Besides, it only supports a single query and requires rebuilding the tree for each new query, rendering it impractical. In contrast, our current noise-adding technique (Lines 9 and 15 of Algorithm 3) utilizes a summation tree such that the error only increases by a factor of $\text{poly} \log n$. This method also supports multiple queries, eliminating the need to rebuild the tree each time.

**How to remove the relative error parameter $\alpha$?** The relative error parameter $\alpha$ in Theorem 3.1 appears because of the $(1+\alpha)$-approximation introduced in Algorithm 5 (Remark G.3) to reduce the number of required iterations from naive $O(n)$ to $O(\log(n)/\alpha)$. However, we notice that a recent work [Liu et al., 2024a] does not utilize $(1 + \alpha)$-approximation and still achieves $O(\log n)$ iteration number. They introduce a new tree node representation where each node stores the sum of distances from one point to multiple points, enabling the answer to be divided into only $\log n$ values, each combining two distance values, two count values, and $y$ itself. Our DPTREE algorithms can be integrated with their method, thus removing parameter $\alpha$.

## B  More Related Work

**Differential Privacy Guarantee Analysis.** Ever since Dwork et al. [2006] proposes the notion of differential privacy (DP), it has become one of the most essential standards of privacy protection in both theoretical and empirical ways [Dwork, 2008, Li et al., 2017, Zhao and Chen, 2022, Ponomareva et al., 2023, Yang et al., 2023]. DP provides a powerful, robust, and quantifiable privacy definition, allowing algorithm design with concrete privacy and accuracy guarantee [Hay et al., 2009, Esfandiari et al., 2022, Andoni et al., 2023, Li and Li, 2023b, Huang and Yi, 2021, Ghazi et al., 2023, Backurs et al., 2024, Cohen-Addad et al., 2022a, Epasto et al., 2024, Chen et al., 2022, Hopkins et al., 2023, Narayanan, 2022, 2023, Jung et al., 2019, Li and Li, 2024, Fan and Li, 2022, Fan et al., 2024, Li and Li, 2023a, Cherapanamjeri et al., 2023, Cohen-Addad et al., 2022b, Dong et al., 2024, Farhadi et al., 2022, Gopi et al., 2021, 2023, Li et al., 2022, Gopi et al., 2022, Eliáš et al., 2020, Song et al., 2023b, Dinur et al., 2023, Woodruff et al., 2023, Song et al., 2023a, Gao et al., 2024]. Additionally, new mechanisms have been proposed beyond the traditional Laplace, Gaussian, and Exponential mechanisms [Dwork and Roth, 2014]. For example, truncated Laplace

mechanism [Geng et al., 2020] is proved to be the current tightest the lower and upper bounds on the minimum noise amplitude and power cross all $(\epsilon, \delta)$-DP distributions.

**Cross-Attention in System Prompt, RAG, Stable Diffusion and More.** Cross-attention [Vaswani et al., 2017], first introduced in language translation, is a widely used technique in many advanced AI systems. For example, Stable Diffusion [Rombach et al., 2022] and SORA [OpenAI, 2024b] employ cross-attention as a core module for a text-to-image conditional generation. This technique is also utilized by other multimodal models [Liang et al., 2024c], including Imagen [Saharia et al., 2022] and Diffusion Transformer [Peebles and Xie, 2023]. In the realm of text-to-image editing, Hertz et al. [2022] analyzes and controls the cross-attention module to enable editing without requiring additional training. Furthermore, Yang et al. [2024] tackles the issue of inaccurate cross-attention maps, enhancing fine-grained control over edited regions while preventing unintended changes to other areas. In addition, Retrieval Augmented Generation (RAG) [Lewis et al., 2020, Borgeaud et al., 2022, Gao et al., 2023], a technique that improves model responses by retrieving information from a knowledge base or external documents, extensively uses cross-attention as its core design module. Cross-attention also has other applications. Oymak et al. [2023] demonstrates that the prompt-tuning [Liang et al., 2024a] task can be formulated as cross-attention, while Chen et al. [2021] uses cross-attention to fuse multi-scale features in vision transformers, thereby reducing computation. Moreover, attention-based Transformer architecture makes LGMs equipping many emergent ability [Wei et al., 2022], such as spatial reasoning [Wang et al., 2024], mathematical reasoning [Li et al., 2024], in-context learning ability [Shi et al., 2024], compositional ability [Xu et al., 2024], few-shot adaptation ability [Shi et al., 2022b, Xu et al., 2023], and so on.

## C  More Preliminary

In Section C.1, we give the probability tools we use in the paper. In Section C.2, we provide the algebraic facts we use. In Section C.3, we give the DP facts we use in the paper. In Section C.4, we compare between popular DP mechanisms.

### C.1  Probability Tools

In this section, we give several probability lemmas.

**Lemma C.1** (Markov's inequality). *If $x$ is a nonnegative random variable and $t > 0$, we have*

$$\Pr[x \geq t] \leq \frac{\mathbb{E}[x]}{t}.$$

**Lemma C.2** (Chernoff bound, [Chernoff, 1952]). *Let $x_i$ be a Bernoulli random variable with probability $p_i$ of being equal to 1 and $1 - p_i$ of being equal to 0, and all $x_i$ for $i \in [n]$ are independent. Let $x = \sum_{i=1}^{n} x_i$. Let $\mu = \mathbb{E}[x] = \sum_{i=1}^{n} p_i$. Then, for all $\delta > 0$ we have*

$$\Pr[x \geq (1 + \delta)\mu] \leq \exp(-\delta^2 \mu/3),$$

*and for all $0 < \delta < 1$*

$$\Pr[x \leq (1 - \delta)\mu] \leq \exp(-\delta^2 \mu/2).$$

**Lemma C.3** (Chebyshev's inequality). *Let $x$ (integrable) be a random variable with finite non-zero variance $\sigma^2$ (and thus finite expected value $\mu$). Then for any real number $k > 0$,*

$$\Pr[|x - \mu| \geq k\sigma] \leq \frac{1}{k^2}.$$

### C.2  Algebraic Facts

**Fact C.4** (Upper bound of exponential, Fact C.9 in Liang et al. [2024b]). *For $a \in \mathbb{R}$, $b \in \mathbb{R}$, $a, b \leq R$, where $R \geq 0$, we have*

$$|\exp(a) - \exp(b)| \leq \exp(R)|a - b|.$$

## C.3 DP Facts

In this section, we present several facts about differential privacy (DP). We first state the post-processing property, which means, in an algorithm, if one step is DP, all the following steps are DP.

**Fact C.5** (Post-processing, see Fact 2.1 in Ghazi et al. [2023]). *Let $\mathcal{A}_1$ be an $(\epsilon, \delta)$-DP algorithm and $\mathcal{A}_2$ be a (randomized) post-processing algorithm. Then the algorithm $\mathcal{A}(X) = \mathcal{A}_2(\mathcal{A}_1(X))$ is still an $(\epsilon, \delta)$-DP algorithm.*

If we have many DP algorithms, we need a composition rule. The most straightforward composition is the basic/sequential composition rule.

**Fact C.6** (Basic composition, see Fact 2.3 in Ghazi et al. [2023]). *Let $\mathcal{A}_1$ be an $(\epsilon_1, \delta_1)$-DP algorithm and $\mathcal{A}_2$ be an $(\epsilon_2, \delta_2)$-DP algorithm. Then $\mathcal{A}(X) = (\mathcal{A}_1(X), \mathcal{A}_2(\mathcal{A}_1(X), X))$ is an $(\epsilon_1 + \epsilon_2, \delta_1 + \delta_2)$-DP algorithm.*

We can do much better if we know that the inputs are disjoint.

**Fact C.7** (Parallel composition, see Fact 2.4 in Ghazi et al. [2023]). *Let $\mathcal{A}_1$ be an $(\epsilon_1, \delta_1)$-DP algorithm and $\mathcal{A}_2$ be an $(\epsilon_2, \delta_2)$-DP algorithm. Assume $\mathcal{A}_1$ and $\mathcal{A}_2$ depend on disjoint subsets of input coordinates. Then the algorithm $\mathcal{A}(X) = (\mathcal{A}_1(X), \mathcal{A}_2(\mathcal{A}_1(X), X))$ is a $(\max\{\epsilon_1, \epsilon_2\}, \max\{\delta_1, \delta_2\})$-DP algorithm.*

In addition, we have the advanced composition, which improves the dependence of the number of DP algorithms to square root but compromises the term $\delta'$.

**Theorem C.8** (Advanced composition, see Theorem 3.20 in Dwork and Roth [2014]). *For all $\epsilon, \delta, \delta' \geq 0$, the class of $(\epsilon, \delta)$-differentially private mechanisms satisfies $(\epsilon', k\delta + \delta')$-differential privacy under $k$-fold adaptive composition for:*

$$\epsilon' = k\epsilon(e^\epsilon - 1) + \epsilon\sqrt{2k \log(1/\delta')}.$$

## C.4 Comparison of Truncated Laplace, Gaussian, and Laplace Mechanisms

We first define the Laplace mechanism as below:

**Definition C.9** (Laplace distribution). *We use $\mathrm{Lap}(b)$ to denote the pdf: $p(z) = \frac{1}{2b} \exp(-\frac{|z|}{b})$.*

**Fact C.10.** *For $z \sim \mathrm{Lap}(b)$, $\mathbb{E}[z] = 0$, and $\mathrm{Var}[z] = 2b^2$. Furthermore, if $b = \Delta/\epsilon$, we have $\mathrm{Var}[z] = 2\Delta^2/\epsilon^2$.*

In this paper, we use the Chebyshev inequality to bound the error, and from Geng et al. [2020], we know that the truncated Laplace mechanism has the current minimum variance across all $(\epsilon, \delta)$-DP distributions.

The variance of Gaussian mechanism in Theorem 3.22 in Dwork and Roth [2014]:

$$\mathrm{Var} = \frac{2\Delta^2 \log(1.25/\delta)}{\epsilon^2}.$$

The variance of Laplace mechanism in Fact C.10:

$$\mathrm{Var} = \frac{2\Delta^2}{\epsilon^2}.$$

The variance of truncated Laplace mechanism in Fact 2.5, for $c \in (0, 1]$:

$$\mathrm{Var} = \frac{2\Delta^2 c}{\epsilon^2}.$$

Thus, since it has the minimum variance, we choose the truncated Laplace mechanism to design our algorithms among these popular mechanisms.

# D   Proof Outline

This section provides the proof outline of our paper. In Section D.1, we analyze our DPTREE data structure. In Section D.2, we show the sensitivity of summation problem. In Section D.3, we explain the high-level idea behind the weighted $\ell_p^p$ distance query. In Section D.4, we show how to answer one-dimensional weighted $\ell_1$ distance query. In Section D.5, we show how to answer Softmax distance query using previous algorithms. In Section D.6, we show how to handle adaptive query. By combining the results from these sections, we prove the main results in Section 4.

## D.1   Summation Segment Tree

First, in order to solve the weighted distance problem, we need to have a basic DP algorithm (Algorithm 3) that can answer simple summation queries. After analyzing its DP and error in Section E, we state the data structure theorem.

**Theorem D.1** (DPTREE data structure, informal version of Theorem E.1). *There is a data structure (see DPTREE in Algorithm 3) that uses $O(n)$ spaces to support the following operations:*

- INIT$(a \in \mathbb{R}^n, n \in \mathbb{N}_+, \Delta \in \mathbb{N}_+, \epsilon \in (0,1), \delta \in (0,1))$. *It takes $O(n)$ time to initialize the data structure.*

- QUERY$(x \in [n], y \in [n])$. *It takes $O(\log n)$ time to output a number $z$ such that*

  - *the process of output $z$ satisfies $(\epsilon, \delta)$-DP private, which computes $\sum_{i=x}^{y} a_i$,*
  - $|z - \sum_{i=x}^{y} a_i| \leq O(\epsilon^{-1}\Delta \log^{3/2} n)$,
  - *it holds with probability $0.99$.*

During the design of the data structure, we found an interesting property based on the parallel composition rule of DP Fact C.7. We will now state the lemma, whose proof is provided in Section E.

**Lemma D.2** (Weighted sum of disjoint interval queries, informal version of Lemma E.8). *If the following conditions hold that:*

- *Let there be $t$ disjoint intervals, i.e., $S_j$ for $j \in [t]$, such that $S_j \cap S_k = \emptyset$ for all $j \neq k$.*

- *Let $\epsilon \in (0,1)$ and $\delta \in (0,1)$.*

- *Let $a_j$ for $j \in [t]$ be a series that square converges to $a$, i.e., $\sum_{j=1}^{t} a_j^2 \leq a$.*

*Then, we have Alg. 3 is $(\epsilon, \delta)$-DP and output $\sum_{j=1}^{t} a_j \text{QUERY}(S_j)$ with the error upper bounded by*

$$O(a^{1/2}\epsilon^{-1}\Delta \log^{3/2} n)$$

*with probability $0.99$.*

From Lemma D.2, we can see that if we have multiple disjoint interval queries, the error of the weighted sum of the intervals can be bounded independently of the number of queries, as long as the sum of squared weights is finite.

## D.2   Sensitivity for Range Summation Problem

Our DP summation tree data structure DPTREE (Algorithm 3) requires sensitivity parameter $\Delta$. In this section, we show that for the summation problem, we have the sensitivity $\Delta = 2R$ if the input $X \in [-R, R]^n$.

**Lemma D.3** (Sensitivity of summation). *Let $X \in [-R, R]^n$. We have the sensitivity $\Delta = 2R$ for* DPTREE.INIT *in Algorithm 3.*

*Proof.* Let's say two neighboring datasets $X$ and $X'$ differ in $x_i$ and $x_i'$ for some $i$ in the array $X$. Then for a summation problem, i.e. $f(X) := \sum_{i=1}^{n} x_i$, we have

$$\Delta = \max_{X,X'} |f(X) - f(X')| = \max_{X,X'} |x_i - x_i'| = 2R.$$

where the first step follows from Definition 2.2, the second step follows from $X, X'$ differ in $x_i, x_i'$, and the last step follows from each coordinate of the dataset is bounded in $[-R, R]$. □

## D.3 Weighted $\ell_p^p$ Distance Problem

In this section, we introduce the intuition behind the method for handling the weighted $\ell_p^p$ distance problem. The formal lemmas and proofs can be found in Section F.

Given a dataset and a query point in $d$ dimensions, we round each coordinate of the data points and the query point to the nearest multiple of a small interval. We then aggregate the weights of data points that have been rounded to the same position. Finally, we compute the sum of these aggregated weights multiplied by the distances between the query point and the data points over the rounded positions. This approach makes the computation more efficient while maintaining sufficient accuracy.

We provide an example of weighted $\ell_1$-distance of a one-dimensional dataset consisting of 10 data points, i.e., $X \in [0,1]^{10}$ and a query $y = 0$ in Figure 1.

**Lemma D.4** (Weighted $\ell_p^p$-distance high dimension, informal version of Lemma F.2). *If the following conditions hold:*

- *Let data $X \in [0,R]^{n \times d}$ and $x_i^\top \in [0,R]^d$ be the $i$-th row of $x$, weight $w \in \mathbb{R}^n$, query $y \in [0,R]^d$.*

- *We round each dimension of $X$ and $y$ to an integer multiple of $R/n$.*

- *Let $x_{i,k}, y_k$ denote the $k$-th coordinates of $x_i, y$ for $k \in [d]$.*

- *Let $c_{j,k} := \sum_{j_0 \in S_{j,k}} w_{j_0}$ where the set $S_{j,k}$ is the set of index $i$ such that the corresponding $x_{i,k}$ is rounded to $jR/n$ for $j \in \{0,1,2,\ldots,n\}$ for $k \in [d]$.*

- *After rounding, we assume that $y_k$ is in the $l_k R/n$ position for $l_k \in \{0,1,2,\ldots,n\}$ for $k \in [d]$.*

*For the weighted problem, we have*

$$\sum_{i=1}^n w_i \cdot \|y - x_i\|_p^p = \sum_{k=1}^d \sum_{j=0}^n (|l_k - j|R/n + O(R/n))^p c_{j,k}.$$

*where $O(R/n)$ is the rounding error for each data point.*

**Remark D.5.** *In Lemma D.4, we first round the dataset. This rounding simplifies the calculation by reducing the number of possible positions to consider, from real values in $[0,R]^d$ to the total $O(nd)$ spots. However, it also introduces an error $O(R/n)$ for one data point. Then, for one spot in the rounded dataset, we sum over the weights of that spot and multiply the corresponding distance raised to the power of $p$. Additionally, since we are dealing with $\ell_p^p$ distance, the rounding error is also raised to the power of $p$.*

## D.4 One-Dimensional Weighted $\ell_1$ Distance Data Structure

Based on previous discussions in Section D.1 and D.3, we can now describe our one-dimensional weighted $\ell_1$ distance data structure, DPTREEDISTANCE, presented in Algorithm 5 and 6, which generalizes the results from Backurs et al. [2024]. Drawing from the intuition in Section D.3, the initialization process is as follows: first, we round each data point in the dataset to the nearest multiple of a small interval and build an array that aggregates the corresponding weights. This array is then fed into our DPTREE data structure in Algorithm 3. At query time, we query the DPTREE to obtain the aggregated weights within a small interval and multiply these weights by the distance to the query point. Furthermore, we also introduce a relative error parameter $\alpha$ to reduce the total number of queries to $O(\log(n)/\alpha)$ instead of querying all $n$ positions. We also analyze the DP and the error bound; see details in Section G.

**Theorem D.6** (DPTREEDISTANCE data structure, informal version of Theorem G.6). *There is a data structure DPTREEDISTANCE (Algorithm 5 and 6) that uses $O(n)$ spaces to solve weighted $\ell_1$-distance query problem for dataset $X \subset [0,R]$ and support the following operations:*

- INIT$(X \subset [0,R], n \in \mathbb{N}_+, w \in [-R_w, R_w]^n, \epsilon \in (0,1), \delta \in (0,1))$. *(Algorithm 5) It takes $O(n)$ time to initialize the data structure.*

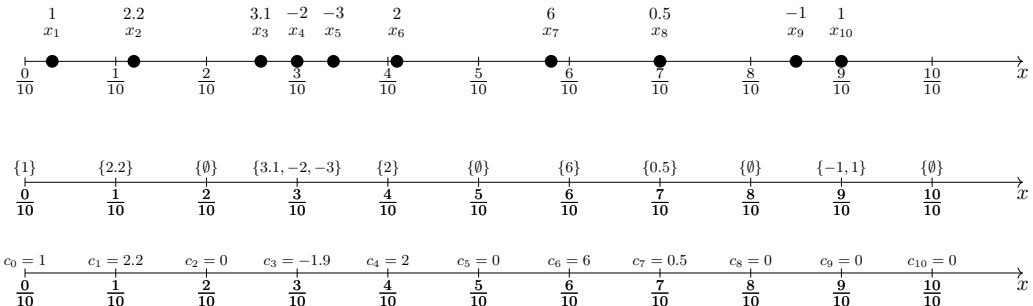

Figure 1: The visualization of how to build $c_j$ of rounded dataset $X \in [0, 1]^{10}$ and compute the weighted $\ell_1$ distance. The number above each $x_i$ is $w_i$. See Algorithm 5 for details. Suppose $y = 0$. Then $\sum_{i=1}^{n} w_i |y - x_i| = 0.1 \cdot 2.2 + 0.3 \cdot 3.1 + 0.3 \cdot (-2) + 0.3 \cdot (-3) + 0.4 \cdot 2 + 0.6 \cdot 6 + 0.7 \cdot 0.5 + 0.9 \cdot (-1) + 0.9 \cdot 1 = 4.4$. And $\sum_{j=0}^{n} |k - j| c_j / n = 0.1 \cdot 2.2 + 0.3 \cdot (-1.9) + 0.4 \cdot 2 + 0.6 \cdot 6 + 0.7 \cdot 0.5 = 4.4$. See details in Lemma F.1.

- DISTANCEQUERY($y \in [0, R], \alpha \in (0, 1)$). *(Algorithm 6) It takes $O(\alpha^{-1} \log^2 n)$ time to output a number $z$ such that*

  - *the process of output $z$ satisfies $(\epsilon, \delta)$-DP, which computes $\sum_{i \in [n]} w_i |y - x_i|$,*
  - $|z - \sum_{i \in [n]} w_i |y - x_i|| \leq \alpha \sum_{i \in [n]} w_i |y - x_i| + O(\epsilon^{-1} \alpha^{-1/2} R R_w \log^{3/2} n)$,
  - *it holds with probability 0.99.*

### D.5 Softmax Activation

We then describe how we extend the previous results to Softmax activation, i.e. exponential inner product function (Definition 4.1). From Alman and Song [2023], we know that Softmax activation can be approximated by polynomial kernel function $P(\cdot)$ with a certain error. The following lemma shows that we can transform weighted Softmax queries into polynomial kernels. More specifically, we have one term that computes the weighted $\ell_2^2$ distance, which is the place where we add DP noises. Because of the decomposability of the $\ell_p^p$ distance, i.e. $\sum_{i \in [n]} w_i \|x_i - y\|_p^p = \sum_{j \in [d]} \sum_{i \in [n]} w_i |x_{i,j} - y_j|^p$, we can easily extend the results of Section D.4 to handle the $\ell_2^2$ distance query. After that, we compute the term for the weighted $\ell_2^2$ norms of approximation kernel exactly. Summing all these terms, with a certain error, we can answer the Softmax query. Related proofs can be found in Section J.

**Lemma D.7** (Weighted Softmax approximation, informal version of Lemma J.6). *Let the accuracy parameter be $\epsilon_s \in (0, 0.1)$. Let $R \geq 1$. Let $r \leq \binom{2s+2d}{2s}$ and $s = O(\max\{\frac{\log(1/\epsilon_s)}{\log(\log(1/\epsilon_s)/R)}, R^2\})$. Let $\Gamma_{R,s} := \max_{j \in [s]} \frac{R^j}{\sqrt{j!}}$ (Definition J.3). Let $P(x) : [0, R]^d \to [0, \Gamma_{R,s}]^r$ be the $s$-th order polynomial kernel function defined in Lemma J.5. Then, we can approximate the exponential inner product using the polynomial kernel function:*

$$w^\top \exp(Xy/d) = -\frac{1}{2} \sum_{j \in [r]} \sum_{i \in [n]} w_i |P(x_i)_j - P(y)_j|^2 + \frac{1}{2} \sum_{i \in [n]} w_i(\|P(x_i)\|_2^2 + \|P(y)\|_2^2)$$
$$+ O(w^\top \exp(Xy/d) \cdot \epsilon_s).$$

*Moreover, the vectors $P(\cdot)$ can be computed in $O(r)$ time.*

### D.6 Adaptive Query

We introduce how we can modify our algorithm to solve the adaptive query problem using some tools in Qin et al. [2022]. Our approach is based on proving that our algorithm can handle any query within the query space with a certain error. Since adaptive queries must lie within this space, our algorithm can effectively handle them. In Section D.5, we demonstrate our algorithm's capability to answer weighted Softmax distance queries with constant probability. We then use the Chernoff

bound to boost the constant probability of our algorithm to a high probability. Next, we apply the notion of an $\epsilon_0$-net to bound all query points within the net using the union bound. Finally, we bound all points in the query space by utilizing the Lipschitz property of the weighted Softmax distance function and introducing an additive error. See the proofs in Sections I and J.

**Lemma D.8** (Adaptive Softmax, informal version of Lemma J.10). *If the following conditions hold:*

- *Let $N$ be the $\ell_\infty$ $\epsilon_0$-net of $\mathcal{B}$, and let $|N|$ be the size of the net $N$.*

- *Let data set $X \in [0, R]^{n \times d}$, weights $w \in [-R_w, R_w]^n$, query $y \in [0, R]^d$.*

- *Let the relative error parameter $\alpha \in (0, 1)$, the failure probability $p_f \in (0, 0.01)$.*

- *We create $l = O(\log((R/\epsilon_0)^r/p_f))$ independent copies of the data structure $\{\text{DPTREESOFTMAX}_j\}_{j=1}^l$ (Algorithm 2) and take the median of the outputs with each data structure instantiated with $(\epsilon/l, (\delta + \delta')/l)$-DP.*

- *Let $f(y) := \text{Median}(\{\text{DPTREESOFTMAX}_j.\text{DISTANCEQUERY}(y, \alpha)\}_{j=1}^l)$.*

- *Let $Z(y) := w^\top \exp(Xy/d)$.*

- *Let $B = O(\epsilon^{-1}\alpha^{-1/2}l\Gamma_{R,s}^2 R_w r \sqrt{\log(l/\delta')} \cdot \log^{3/2} n)$.*

*Then with probability $1 - p_f$, for all query points $q \in \mathcal{B}$, there exists a point $y \in N$ which is the closest to $q$, we can have the process of outputting median of $l$ responses is $(\epsilon, \delta + \delta')$-DP and the error satisfies*

$$|f(y) - Z(q)| \leq (\alpha + \epsilon_s)Z(q) + B + 2n\sqrt{d}RR_w \exp(R^2)\epsilon_0.$$

## E   DPTree Algorithm

In this section, we give the analysis of privacy, accuracy and runtime of our DPTREE (Algorithm 3). In Section E.1, we give the theorem (Theorem E.1) of our data structure that can answer summation problem. In Section E.2, we improve our data structure from constant probability to high probability by applying Chernoff bound. In Section E.3, we give the analysis. In Section E.4, we show some results of our data structure if the input queries are disjoint.

### E.1   Single Data Structure

We give the theorem of our DPTREE data structure that can answer the summation problem with DP, accuracy, runtime guarantee.

**Theorem E.1** (DPTREE data structure, formal version of Theorem D.1). *There is a data structure (see DPTREE in Algorithm 3) that uses $O(n)$ spaces to support the following operations:*

- *INIT($a \in \mathbb{R}^n, n \in \mathbb{N}_+, \Delta \in \mathbb{N}_+, \epsilon \in (0, 1), \delta \in (0, 1)$). It takes $O(n)$ time to initialize the data structure.*

- *QUERY($x \in [n], y \in [n]$). It takes $O(\log n)$ time to output a number $z$ such that*

  - *the process of output $z$ satisfies $(\epsilon, \delta)$-DP private, which computes $\sum_{i=x}^y a_i$,*
  - *$|z - \sum_{i=x}^y a_i| \leq O(\epsilon^{-1}\Delta \log^{3/2} n)$,*
  - *it holds with probability 0.99.*

*Proof.* The proofs follow from combining Lemma E.3 (running time of initialization), Lemma E.4 (running time of query), Lemma E.5 (DP of query), and Lemma E.6 (error of query) together. □

---

**Algorithm 3** DPTree initialization and query

---

1: **datastructure** DPTREE                                          ▷ Theorem D.1
2: **members**
3:     $b : \mathbb{R}^{2n-1}$
4:     $c : \mathbb{R}^{2n-1}$
5: **end members**
6: **procedure** INIT($a \in \mathbb{R}^n, n \in \mathbb{N}_+, \Delta \in \mathbb{R}, \epsilon \in (0,1), \delta \in (0,1)$)     ▷ Lemma E.3, Lemma D.3
7:     $b[n, 2n-1] \leftarrow a$
8:     **for** $i = n \rightarrow 2n - 1$ **do**
9:         $c[i] \leftarrow b[i] + \text{TLap}(\Delta, \epsilon/\log n, \delta/\log n)$
10:     **end for**
11:     **for** $i = (\log n) \rightarrow 1$ **do**
12:         **for** $j = 1 \rightarrow 2^{i-1}$ **do**
13:             $k \leftarrow 2^{i-1} + j - 1$
14:             $b[k] \leftarrow b[2k] + b[2k+1]$
15:             $c[k] \leftarrow b[k] + \text{TLap}(\Delta, \epsilon/\log n, \delta/\log n)$
16:         **end for**
17:     **end for**
18: **end procedure**
19: **procedure** QUERY($x \in [n], y \in [n]$)                          ▷ Lemma E.4, E.5, E.6
20:     Trace from bottom nodes of $x$ and $y$ to find their lowest common ancestor, then we report the summation (based on $c$) by using at most $2 \log n$ nodes on the path. Let Value be the above summation.
21:     **return** Value
22: **end procedure**
23: **procedure** TRUEQUERY($x \in [n], y \in [n]$)
24:     Trace from bottom nodes of $x$ and $y$ to find their lowest common ancestor, then we report the summation (based on $b$) by using at most $2 \log n$ nodes on the path, where the height of the tree is $\log n$, and we need left and right boundary points. Let Value be the above summation.
25:     **return** Value
26: **end procedure**
27: **end datastructure**

---

## E.2 Boost the Constant Probability to High Probability

We can use Chernoff bound to boost the high probability by repeating the data structure multiple times.

**Theorem E.2** (High-probability). *There is a data structure (see* DPTREEHIGHPROB *in Algorithm 4) that uses* $O(n \log(1/\delta_{\text{fail}}))$ *spaces to support the following operations*

- INIT($a \in \mathbb{R}^n, n \in \mathbb{N}_+, \Delta \in \mathbb{N}_+, \epsilon \in (0,1), \delta \in (0,1), \delta_{\text{fail}} \in (0, 0.01)$). *It takes* $O(n \log(1/\delta_{\text{fail}}))$ *time to initialize the data structure.*

- QUERY($x \in [n], y \in [n]$). *It takes* $O(\log(n) \cdot \log(1/\delta_{\text{fail}}))$ *time to output a number* $z$ *such that*

    - *the process of output* $z$ *satisfies* $(\epsilon, \delta)$-*DP private, which computes* $\sum_{i=x}^{y} a_i$,
    - $|z - \sum_{i=x}^{y} a_i| \leq O(\epsilon^{-1} \Delta \log^{3/2}(n) \cdot \log(1/\delta_{\text{fail}}))$,
    - *it holds with probability* $1 - \delta_{\text{fail}}$ *for failure probability* $\delta_{\text{fail}} \in (0, 0.01)$.

*Proof.* Note that our data structure (Theorem E.1) succeeds with probability 0.99. The success of the algorithm (Theorem E.1) can be viewed as a Bernoulli random variable, to which we apply the Chernoff bound (Lemma C.2). By repeating the data structure $O(\log(1/\delta_{\text{fail}}))$ times and taking the median of the outputs, we boost the success probability. The details are following.

To boost the success probability, we assume the query is repeated $l$ times. Let $i \in [l]$, and let $z_i$ denote the indicator random variable for the success of the $i$-th instance of the data structure for a single query. Let $z = \sum_{i=1}^{l} z_i$ be the total success times. Since $p = \Pr[z_i = 1] = 0.99$, we can

have $\mu = \mathbb{E}[z] = \sum_{i=1}^{l} p = lp$. Note that $p = 0.99$. By setting $\delta = 0.1$ and using Chernoff bound from Lemma C.2, we can show

$$\Pr[z \leq l/2] \leq \Pr[z \leq (1 - \delta)lp] \leq \exp(-\delta^2 lp/2).$$

Note that we want $z > l/2$ (since we want at least half to succeed so we could take the median),

$$\Pr[z > l/2] \geq 1 - \exp(-\delta^2 lp/2).$$

To ensure that failure probability is $\delta_{\text{fail}}$, we have

$$\exp(-\delta^2 lp/2) = \delta_{\text{fail}}.$$

We can make this hold by choosing $l = O(\log(1/\delta_{\text{fail}}))$.

By the DP basic composition rule (Fact C.6), we need to choose $\epsilon = \epsilon'/O(\log(1/\delta_{\text{fail}}))$ and $\delta = \delta'/O(\log(1/\delta_{\text{fail}}))$ where $\epsilon', \delta'$ are the $\epsilon, \delta$ in Theorem E.1. $\qquad\square$

---

**Algorithm 4** Boost constant probability

1: **datastructure** DPTREEHIGHPROB                                                      ▷ Theorem E.2
2: **members**
3:     $\mathcal{D}_1, \ldots, \mathcal{D}_{O(\log(1/\delta_{\text{fail}}))}$ : DPTREE                              ▷ Alg. 3
4: **end members**
5: **procedure** INIT($a \in \mathbb{R}^n, n \in \mathbb{N}_+, \Delta \in \mathbb{N}_+, \epsilon \in (0, 1), \delta \in (0, 1), \delta_{\text{fail}} \in (0, 0.01)$)
6:     **for** $i = 1 \rightarrow O(\log(1/\delta_{\text{fail}}))$ **do**
7:         $\mathcal{D}_i$.INIT($a, n, \Delta, \epsilon/O(\log(1/\delta_{\text{fail}})), \delta/O(\log(1/\delta_{\text{fail}}))$)
8:     **end for**
9: **end procedure**
10: **procedure** QUERY($x \in [n], y \in [n]$)
11:     $r \leftarrow 0^{O(\log(1/\delta_{\text{fail}}))}$
12:     **for** $i = 1 \rightarrow O(\log(1/\delta_{\text{fail}}))$ **do**
13:         $r_i \leftarrow \mathcal{D}_i$.QUERY($x, y$)
14:     **end for**
15:     **return** Median of $r$
16: **end procedure**
17: **end datastructure**

---

### E.3 Algorithm of Data Structure

In this section, we analyze the accuracy, DP, and runtime of Algorithm 3.

We first analyze the runtime.

**Lemma E.3** (Runtime of initialization, Algorithm 3). *For the initialization, we have the time complexity of Algorithm 3 is $O(n)$.*

*Proof.* All the computations are dominated by $O(n)$ time. $\qquad\square$

**Lemma E.4** (Runtime of query, Algorithm 3). *For each query, we have the time complexity of Algorithm 3 is $O(\log n)$.*

*Proof.* Due to the property of tree, we will use at most $2 \log n$ nodes in the tree, thus the running time is $O(\log n)$. $\qquad\square$

We now analyze the DP.

**Lemma E.5** (Privacy of query, Algorithm 3). *The output process of* QUERY *(see Algorithm 3) is $(\epsilon, \delta)$-DP.*

*Proof.* Suppose that our dataset is $X \in [-R, R]^n$. Note that we only add noise in the pre-processing stage. There is no noise in the query stage. Since the problem we care about is summation, if we change one leaf node, the sensitivity $\Delta = 2R$ (see Lemma D.3). Since we add noise to each node in the tree, and each leaf node count will contribute to $\log n$ nodes, it is equivalent to our output function being in $\log n$ dimension. We will then blow up the DP parameter by $\log n$ factor. Thus, using the basic composition rule (Fact C.6), the DP guarantee for the whole tree data structure is $((\epsilon/\log n) \cdot \log n, (\delta/\log n) \cdot \log n)$ which is $(\epsilon, \delta)$-DP. □

We now analyze the accuracy.

**Lemma E.6** (Accuracy of query, Algorithm 3). *Let $\epsilon \in (0,1)$ and $\delta \in (0,1)$. Then, using Chebyshev's inequality and Fact 2.5, we have the error of* QUERY*(see Algorithm 3) output is upper bounded by:*

$$O(\epsilon^{-1}\Delta \log^{3/2} n).$$

*with probability* 0.99.

*Proof.* For an interval $S$, we define TRUEQUERY$(S)$ to be the output of DPTREE.TRUEQUERY in Algorithm 3. Let QUERY$(S)$ denote the noised interval query answer returned by DPTREE.QUERY in Algorithm 3. Let $z := $ QUERY$(S) - $ TRUEQUERY$(S)$, which from Algorithm 3 we can see this is the sum of $O(\log n)$ independent truncated Laplace random variables each with parameter TLap$(\Delta, \epsilon/\log n, \delta/\log n)$. Thus,

$$z = \sum_{i=1}^{O(\log n)} z_i$$

where $z_i \sim $ TLap$(\Delta, \epsilon/\log n, \delta/\log n)$, and every $z_i$ are independent to each other.

We know $\mu = \mathbb{E}[z] = 0$ since $\mathbb{E}[z_i] = 0$. From Fact 2.5, we know the variance for each $z_i$ is Var$[z_i] = c\epsilon^{-2}\Delta^2 \log^2 n$ where $0 < c \leq 2$ and $c = 2$ when $\delta = 0$.

Therefore, we can show

$$\begin{aligned}
\mathrm{Var}[z] &= \mathrm{Var}\Big[ \sum_{i=1}^{O(\log n)} z_i \Big] \\
&= \sum_{i=1}^{O(\log n)} \mathrm{Var}[z_i] \\
&= O(c\epsilon^{-2}\Delta^2 \log^3 n)
\end{aligned} \tag{4}$$

where the first step follows from definition of $z$, the second step follows from every $z_i$ are independent to each other, and the last step follows from $\mathrm{Var}[z_i] = O(c\epsilon^{-2}\Delta^2 \log^2 n)$.

Note that we wish to bound $|z| = |$QUERY$(S) - $TRUEQUERY$(S)|$ as our error.

Using Lemma C.3, we can have

$$\Pr[|z| \geq k\sigma] \leq \frac{1}{k^2}.$$

We know that $\sigma = \sqrt{\mathrm{Var}[z]} = O(c^{1/2}\epsilon^{-1}\Delta \log^{3/2} n)$. Picking $k = 10$, we have

$$\Pr[|z| < 10\sigma] \geq 0.99.$$

Thus, we conclude that error is bounded by $O(c^{1/2}\epsilon^{-1}\Delta \log^{3/2} n) = O(\epsilon^{-1}\Delta \log^{3/2} n)$ (since $c \in (0, 2]$) with probability 0.99. □

### E.4 Disjoint Intervals

In this section, we show some interesting results for our DPTREE data structure if the input queries are disjoint.

**Lemma E.7** (Sum of disjoint interval queries). *If the following conditions hold that:*

- *Let there be $t$ disjoint intervals, i.e., $S_j$ for $j \in [t]$, such that $S_j \cap S_k = \emptyset$ for all $j \neq k$.*

- *Let $\epsilon \in (0, 1)$ and $\delta \in (0, 1)$.*

*Then, we have Algorithm 3 is $(\epsilon, \delta)$-DP and outputs $\sum_{j=1}^{t} \mathrm{QUERY}(S_j)$ with the error upper bounded by*

$$O(t^{1/2}\epsilon^{-1}\Delta \log^{3/2} n)$$

*with probability* $0.99$.

*Proof.* From Lemma E.5, we know that DPTree.QUERY is $(\epsilon, \delta)$-DP. Then, from Fact C.7 and the disjoint intervals in Algorithm 6, we can conclude that the value returned is $(\epsilon, \delta)$-DP.

Let $\mathrm{TRUEQUERY}(S_j)$ denote the true interval query answer returned by DPTREE.TRUEQUERY in Algorithm 3 for interval $S_j$. Let $\mathrm{QUERY}(S_j)$ denote the noised interval query answer returned by DPTREE.QUERY in Algorithm 3 for interval $S_j$. Let $z_j := \mathrm{QUERY}(S_j) - \mathrm{TRUEQUERY}(S_j)$ and $z = \sum_{j=1}^{t} z_j$. From the proof of Lemma E.6, we know $z_j$ is the sum of $O(\log n)$ independent truncated Laplace random variables each with parameter $\mathrm{TLap}(\Delta, \epsilon/\log n, \delta/\log n)$ and the variance is bounded by

$$\mathrm{Var}[z_j] = O(\epsilon^{-2}\Delta^2 \log^3 n)$$

Since the intervals $S_j$ are disjoint, they are independent to each other. Then, we have

$$\mathrm{Var}[z] = \mathrm{Var}\left[\sum_{j=1}^{t} z_j\right]$$
$$= \sum_{j=1}^{t} \mathrm{Var}[z_j]$$
$$= O(t\epsilon^{-2}\Delta^2 \log^3 n)$$

where the first step follows from definition of $z$, the second step follows from the intervals are disjoint, and the last step follows from $\mathrm{Var}[z_j] = O(\epsilon^{-2}\Delta^2 \log^2 n)$.

Note that we wish to bound $|z|$ as our error.

Using Lemma C.3, we can have error bounded by

$$O(t^{1/2}\epsilon^{-1}\Delta \log^{3/2} n)$$

with probability $0.99$. $\qquad\square$

Moreover, this can be generalized to weighted sum of queries.

**Lemma E.8** (Weighted sum of disjoint interval queries, formal version of Lemma D.2). *If the following conditions hold that:*

- *Let there be $t$ disjoint intervals, i.e., $S_j$ for $j \in [t]$, such that $S_j \cap S_k = \emptyset$ for all $j \neq k$.*

- *Let $\epsilon \in (0, 1)$ and $\delta \in (0, 1)$.*

- *Let $a_j$ for $j \in [t]$ be a series that square converges to $a$, i.e., $\sum_{j=1}^{t} a_j^2 \leq a$.*

*Then, we have Alg. 3 is $(\epsilon, \delta)$-DP and output $\sum_{j=1}^{t} a_j\mathrm{QUERY}(S_j)$ with the error upper bounded by*

$$O(a^{1/2}\epsilon^{-1}\Delta \log^{3/2} n)$$

*with probability* $0.99$.

*Proof.* The DP proof is the same as in the proof of Lemma E.7.

Let $\text{TRUEQUERY}(S_j)$ and $\text{QUERY}(S_j)$ be same in the proof of Lemma E.7 Let $z_j := \text{QUERY}(S_j) - \text{TRUEQUERY}(S_j)$ and $z = \sum_{j=1}^{t} a_j z_j$. From the proof of Lemma E.7, we know the variance of $z_j$ is bounded by

$$\text{Var}[z_j] = O(\epsilon^{-2} \Delta^2 \log^3 n)$$

Since the intervals $S_j$ are disjoint, they are independent to each other. Then, we have

$$\begin{aligned}
\text{Var}[z] &= \text{Var}[\sum_{j=1}^{t} a_j z_j] \\
&= \sum_{j=1}^{t} \text{Var}[a_j z_j] \\
&= \sum_{j=1}^{t} a_j^2 \text{Var}[z_j] \\
&= \sum_{j=1}^{t} a_j^2 \cdot O(\epsilon^{-2} \Delta^2 \log^3 n) \\
&= O(a \epsilon^{-2} \Delta^2 \log^3 n)
\end{aligned}$$

where the first step follows from the definition of $z$, the second step follows from the intervals are disjoint, the third step follows from the $\text{Var}[az] = a^2 \text{Var}[z]$ for a random variable $z$ and a constant $a$, the fourth step follows from the $\text{Var}[z_j] = O(\epsilon^{-2} \Delta^2 \log^2 n)$, and the last step follows from $\sum_{j=1}^{t} a_j^2 \leq a$.

Note that we wish to bound $|z|$ as our error.

Using Lemma C.3, we can have error bounded by

$$O(a^{1/2} \epsilon^{-1} \Delta \log^{3/2} n)$$

with probability 0.99. $\qquad \square$

# F   Weighted $\ell_p^p$ Distance

In this section, we introduce how to handle weighted $\ell_p^p$ distance problem in the high level idea. In Section F.1, we show how to solve one dimensional weighted problem. In Section F.2, we show how to solve high dimensional weighted problem by decomposing each coordinate of the high dimensional dataset.

Suppose we have the original data $X \in [0, R]^n$ and weight $w \in \mathbb{R}^n$ and query $y \in [0, R]$. We want to compute the weighted $\ell_1$-distance, i.e.

$$\sum_{i=1}^{n} w_i \cdot |y - x_i|.$$

For data in $d$-dimension, due to the decomposability of $\ell_p^p$ distance, our problem will be: given $x_i \in [0, R]^d$ and $w_i \in \mathbb{R}$ for $i \in [n]$, and $y \in [0, R]^d$, we can compute

$$\sum_{i=1}^{n} w_i \cdot \|y - x_i\|_p^p = \sum_{j=1}^{d} \sum_{i=1}^{n} w_i \cdot |y_j - x_{i,j}|^p$$

where $x_{i,j}, y_j$ means the $j$-th coordinates of $x_i, y$ for $j \in [d]$.

Therefore, we can solve one dimension problem first, and then the high dimension case can be solved automatically.

### F.1 One Dimensional Weighted Distance

Now we can give the lemma for weighted distance of dataset.

**Lemma F.1** (Weighted distance one dimension). *If the following conditions hold:*

- *Let data $X \in [0, R]^n$, weight $w \in \mathbb{R}^n$, query $y \in [0, R]$.*

- *We round $X$ and $y$ to an integer multiple of $R/n$.*

- *Let $c_j = \sum_{j_0 \in S_j} w_{j_0}$ where set $S_j$ is the set of index $i$ such that the corresponding $x_i$ is rounded to $jR/n$ for $j \in \{0, 1, 2, \ldots, n\}$.*

- *After rounding, we assume $y$ is in the $kR/n$ position for $k \in \{0, 1, 2, \ldots, n\}$.*

*For the weighted problem, we have*

$$\sum_{i=1}^{n} w_i \cdot |y - x_i| = \sum_{j=0}^{n} (|k - j|R/n + O(R/n))c_j.$$

*Moreover, we have*

$$\sum_{i=1}^{n} w_i \cdot |y - x_i|^p = \sum_{j=0}^{n} (|k - j|R/n + O(R/n))^p c_j$$

*where $O(R/n)$ is the rounding error for each data point.*

*Proof.* For each $i$, we have:

$$w_i \cdot |y - x_i| = w_i \cdot (\frac{|k - j|R}{n} + O(\frac{R}{n})).$$

where $O(R/n)$ is the rounding error introduced by each data point, since each data point will be at most $O(R/n)$ away from its true position.

We can construct $c_j$ by

$$c_j = \sum_{j_0 \in S_j} w_{j_0}$$

set $S_j$ is the set of index $i$ such that the corresponding $x_i$ is rounded to $jR/n$. Moreover, $c_j$ can be negative.

Summing over all $i$ and grouping by $j$, we get:

$$\sum_{i=1}^{n} w_i \cdot |y - x_i| = \sum_{j=0}^{n} (\frac{|k - j|R}{n} + O(\frac{R}{n}))c_j.$$

The total rounding error will be $O(R)$ because we have $n$ data points, each with an error of at most $O(R/n)$.

Moreover, we have

$$\sum_{i=1}^{n} w_i \cdot |y - x_i|^p = \sum_{j=0}^{n} (\frac{|k - j|R}{n} + O(\frac{R}{n}))^p c_j.$$

$\square$

### F.2 High Dimensional Weighted Distance

Finally, we can solve the problem of weighted distance for $d$-dimensional dataset.

**Lemma F.2** (Weighted $\ell_p^p$-distance high dimension, formal version of Lemma D.4). *If the following conditions hold:*

- *Let data $X \in [0, R]^{n \times d}$ and $x_i^\top \in [0, R]^d$ be the i-th row of x, weight $w \in \mathbb{R}^n$, query $y \in [0, R]^d$.*

- *We round each dimension of X and y to an integer multiple of $R/n$.*

- *Let $x_{i,k}, y_k$ denote the k-th coordinates of $x_i, y$ for $k \in [d]$.*

- *Let $c_{j,k} := \sum_{j_0 \in S_{j,k}} w_{j_0}$ where set $S_{j,k}$ is the set of index i such that the corresponding $x_{i,k}$ is rounded to $jR/n$ for $j \in \{0, 1, 2, \ldots, n\}$ for $k \in [d]$.*

- *After rounding, we assume $y_k$ is in the $l_k R/n$ position for $l_k \in \{0, 1, 2, \ldots, n\}$ for $k \in [d]$.*

*For the weighted problem, we have*

$$\sum_{i=1}^n w_i \cdot \|y - x_i\|_p^p = \sum_{k=1}^d \sum_{j=0}^n (|l_k - j|R/n + O(R/n))^p c_{j,k}$$

*where $O(R/n)$ is the rounding error for each data point.*

*Proof.* We can show

$$\sum_{i=1}^n w_i \cdot \|y - x_i\|_p^p = \sum_{k=1}^d \sum_{i=1}^n w_i \cdot |y_k - x_{i,k}|^p$$

$$= \sum_{k=1}^d \sum_{j=0}^n (|l_k - j|R/n + O(R/n))^p c_{j,k}$$

where the first step follows from decomposability of $\ell_p^p$-distance by dimension, the second step follows from Lemma F.1.

$\square$

# G   One-Dimensional Weighted $\ell_1$ Distance Query

In this section, we generalize the algorithms in Backurs et al. [2024] to weighted distance. Here, we compute the problem of one-dimensional weighted $\ell_1$ distance query i.e. $\sum_{i \in [n]} w_i |y - x_i|$ for a given query $y \in [0, R]$, weights $w \in [-R_w, R_w]^n$ and dataset $X \subset [0, R]$ and $n = |X|$. In Section G.1, we analyze the runtime of our algorithm. In Section G.2, we analyze the DP and accuracy of our algorithm. In Section G.3, we give the theorem for our DPTREEDISTANCE data structure.

## G.1   Runtime Analysis

We first analyze the runtime.

**Lemma G.1** (Runtime of initialization, Algorithm 5). *For the initialization, we have the time complexity of* INIT *in Algorithm 5 is $O(n)$.*

*Proof.* In the initialization of INIT, the computations need $O(n)$ time to compute the count and $O(\log n)$ time to build the tree. Thus, total time is $O(n)$. $\square$

**Lemma G.2** (Runtime of DISTANCEQUERY, Algorithm 6). *For the $\ell_1$ distance query, we have the time complexity of* DISTANCEQUERY *in Algorithm 6 is $O(\alpha^{-1} \log^2 n)$.*

*Proof.* In DISTANCEQUERY, the computations need $O(\log n)$ time to compute one value from DP-TREE.QUERY and this process need to be repeated $O(\alpha^{-1} \log n)$ times. $\square$

**Remark G.3.** *In Line 8 and 13 of Algorithm 6, we use $R/(1 + \alpha)^j$ to approximate the distance of each data point to the query in Lemma F.1, i.e. $|k - j|R/n$. This will introduce $\alpha$ relative error but also reduce the numbers of iteration from $O(n)$ to $O(\log(n)/\alpha)$.*

---

**Algorithm 5** Pre-processing data structure

---

1: **datastructure** DPTREEDISTANCE                                              ▷ Theorem G.6
2: **members**
3:     $\mathcal{D}$ : DPTREE                                                      ▷ Alg. 3
4:     $X : [0, R]^n$
5:     $w : [-R_w, R_w]^n$
6: **end members**
7: **procedure** INIT($X \subset [0, R], n \in \mathbb{N}_+, w \in [-R_w, R_w]^n, \epsilon \in (0, 1), \delta \in (0, 1)$ )   ▷ Lemma F.1
8:     $X, w, a \leftarrow X, w, 0^{n+1}$
9:     **for** $i = 1 \rightarrow n$ **do**
10:        $j \leftarrow \text{ROUND}(x_i, n)$                                     ▷ $x_i \in X$ for $i \in [n]$
11:        $a_j \leftarrow a_j + w_i$
12:    **end for**
13:    $\mathcal{D}.\text{INIT}(a, n + 1, 2R_w, \epsilon, \delta)$                 ▷ Alg. 3, Lemma D.3
14: **end procedure**
15: **procedure** ROUND($x \in [0, R], n \in \mathbb{N}_+$)
16:    Let $j \in \{0, 1, 2, \dots n - 1\}$ denote the integer such that $jR/n \leq x < (j + 1)R/n$
17:    **if** $|x - (j + 1)R/n| \leq |x - jR/n|$ **then**
18:        $j \leftarrow j + 1$
19:    **end if**
20:    **return** $j$
21: **end procedure**
22: **end datastructure**

---

---

**Algorithm 6** One dimensional weighted $\ell_1$ distance query

---

1: **datastructure** DPTREEDISTANCE                                              ▷ Theorem G.6
2: **procedure** DISTANCEQUERY($y \in [0, R], \alpha \in (0, 1)$)  ▷ Lemma G.2, Lemma G.4, Lemma G.5
3:     $y \leftarrow \text{ROUND}(y, n) \cdot (R/n)$                              ▷ Alg. 5
4:     Value $\leftarrow 0$
5:     **for** $j = 0, 1, ..., O(\log(n)/\alpha)$ **do**
6:         $l_j \leftarrow \text{ROUND}(y + \frac{R}{(1+\alpha)^{j+1}}, n)$
7:         $r_j \leftarrow \text{ROUND}(y + \frac{R}{(1+\alpha)^j}, n)$           ▷ Consider the points to the right of $y$
8:         Value $\leftarrow$ Value $+ \mathcal{D}.\text{QUERY}(l_j, r_j) \cdot \frac{R}{(1+\alpha)^j}$   ▷ Alg. 3
9:     **end for**
10:    **for** $j = 0, 1, ..., O(\log(n)/\alpha)$ **do**
11:        $l_j \leftarrow \text{ROUND}(y - \frac{R}{(1+\alpha)^j}, n)$
12:        $r_j \leftarrow \text{ROUND}(y - \frac{R}{(1+\alpha)^{j+1}}, n)$       ▷ Consider the points to the left of $y$
13:        Value $\leftarrow$ Value $+ \mathcal{D}.\text{QUERY}(l_j, r_j) \cdot \frac{R}{(1+\alpha)^j}$   ▷ Alg. 3
14:    **end for**
15:    **Return** Value
16: **end procedure**
17: **end datastructure**

---

### G.2  Privacy and Accuracy Analysis

We show the DP.

**Lemma G.4** (Privacy of DISTANCEQUERY, Algorithm 6)**.** *The output process of* DISTANCE-QUERY *(Algorithm 6) is* $(\epsilon, \delta)$-*DP.*

*Proof.* From Lemma E.5, we know that DPTree.QUERY is $(\epsilon, \delta)$-DP output. We observe that intervals in Algorithm 6 are disjoint. Then, following the same logic in the proof of Lemma E.8, we can conclude that the value returned is $(\epsilon, \delta)$-DP.  □

We now analyze the accuracy of the algorithm.

**Lemma G.5** (Accuracy of DISTANCEQUERY, Algorithm 6). *If the following conditions are satisfied:*

- *Let $X \in [0, R]^n$ be a dataset consisting of $n$ one-dimensional numbers, with weights $w \in [-R_w, R_w]^n$.*

- *Let $\alpha \in (0, 1)$ represent the relative error parameter utilized in Algorithm 6.*

- *Let $\widetilde{A}$ denote the output of the DISTANCEQUERY in Algorithm 6.*

- *Let $A_* := \sum_{i \in [n]} w_i |y - x_i|$ represent the true distance query value for a specific query $y$.*

*Then with probability $0.99$, we have*

$$|\widetilde{A} - A_*| \leq \alpha A_* + O(\epsilon^{-1} \alpha^{-1/2} R R_w \log^{3/2} n).$$

*Proof.* To simplify the explanation, we consider only the distance query for the points in $X$ located to the right of $y$. The proof can be symmetrically applied to the case of points to the left of $y$. For an interval $S_j := (l_j, r_j)$ where $l_j, r_j$ are defined in Algorithm 6, we define TRUEQUERY$(S_j)$ to be the output of DPTREE.TRUEQUERY in Algorithm 3. Let

$$\widehat{A} := \sum_{j=0}^{O(\log(n)/\alpha)} \frac{R}{(1 + \alpha)^j} \cdot \text{TRUEQUERY}(S_j).$$

Since TRUEQUERY returns the sum of the corresponding weights, it aligns with the true answer $A_* := \sum_{i \in [n]} w_i |y - x_i|$. Thus, we have

$$|\widehat{A} - A_*| \leq \alpha \cdot A_*,$$

because for all $j$, the distances between $y$ and different points in $X$ vary only by a multiplicative factor of $(1 + \alpha)$.

Next we show the additive error. Let QUERY$(S_j)$ denote the noised interval query answer returned by DPTREE.QUERY in Algorithm 3. Algorithm 6 outputs $\widetilde{A} = \sum_{j=0}^{O(\log(n)/\alpha)} \frac{R}{(1+\alpha)^j} \cdot \text{QUERY}(S_j)$. We wish to bound

$$|\widehat{A} - \widetilde{A}| \leq \left| \sum_{j=0}^{O(\log(n)/\alpha)} \frac{R}{(1 + \alpha)^j} \cdot (\text{TRUEQUERY}(S_j) - \text{QUERY}(S_j)) \right|.$$

Let $z_j := \text{QUERY}(S_j) - \text{TRUEQUERY}(S_j)$, which from Algorithm 3 we can see this is the sum of $O(\log n)$ independent truncated Laplace random variables.

From Lemma E.8, we only need to show that the series $\frac{1}{(1+\alpha)^j}$ for $j \in \{0, 1, \ldots, O(\log(n)/\alpha)\}$ square converges to $1/\alpha$, since $R$ is a constant.

We can show

$$\sum_{j=0}^{O(\log(n)/\alpha)} \frac{1}{(1 + \alpha)^{2j}} \leq \sum_{j=0}^{\infty} \frac{1}{(1 + \alpha)^{2j}}$$

$$\leq \sum_{j=0}^{\infty} \frac{1}{(1 + \alpha)^j}$$

$$= \frac{1}{1 - \frac{1}{1+\alpha}}$$

$$= 1 + \frac{1}{\alpha}$$

$$= O(1/\alpha)$$

where the first step follows from we extend the finite sum to infinite sum, the second step follows from $\frac{1}{(1+\alpha)^{2j}} \leq \frac{1}{(1+\alpha)^j}$, the third step follows from the closed form of geometric sum, the fourth step follows from simple algebra, and the last step follows from $\alpha \in (0, 1)$.

Then from the proof of Lemma E.8, we can know that the variance is given by

$$O(\frac{R^2 R_w^2 \log^3 n}{\alpha \epsilon^2}) \tag{5}$$

since the sensitivity $\Delta = 2R_w$ from Lemma D.3.

Using Lemma C.3, we can have additive error bounded by

$$O(\frac{R \cdot R_w \log^{3/2} n}{\epsilon \sqrt{\alpha}}).$$

with probability 0.99. $\qquad\qquad\square$

## G.3 One Dimension Single Data Structure

We therefore have the data structure that can solve weighted $\ell_1$-distance problem.

**Theorem G.6** (DPTREEDISTANCE data structure, formal version of Theorem D.6)**.** *There is a data structure* DPTREEDISTANCE *(Algorithm 5,6) that uses $O(n)$ spaces to solve weighted $\ell_1$-distance query problem for dataset $X \subset [0, R]$ and support the following operations:*

- INIT($X \subset [0, R], n \in \mathbb{N}_+, w \in [-R_w, R_w]^n, \epsilon \in (0, 1), \delta \in (0, 1)$). *(Algorithm 5) It takes $O(n)$ time to initialize the data structure.*

- DISTANCEQUERY($y \in [0, R], \alpha \in (0, 1)$). *(Algorithm 6) It takes $O(\alpha^{-1} \log^2 n)$ time to output a number $z$ such that*

   - *the process of output $z$ satisfies $(\epsilon, \delta)$-DP private, which computes $\sum_{i \in [n]} w_i |y - x_i|$,*

   - *$|z - \sum_{i \in [n]} w_i |y - x_i|| \leq \alpha \sum_{i \in [n]} w_i |y - x_i| + O(\epsilon^{-1} \alpha^{-1/2} R R_w \log^{3/2} n)$,*

   - *it holds with probability 0.99.*

*Proof.* The proofs follow from combining Lemma G.1 (running time of initialization), Lemma G.2 (running time of query), Lemma G.4 (DP of query), and Lemma G.5 (error of query) together. $\quad\square$

## H High-Dimensional Weighted $\ell_1$ Query

In this section, we show how we can solve the high dimensional weighted $\ell_1$ distance problem, generalizing results from Backurs et al. [2024]. In Section H.1, we give the analysis of Algorithm 7. In Section H.2, we give the theorem of our DPTREEHIGHDIM data structure.

Algorithm 5,6 can be naturally extended to higher dimensions because of the decomposability of the $\ell_1$ distance function. We construct $d$ separate one-dimensional distance query data structures, each corresponding to a coordinate projection of the dataset.

### H.1 Privacy and Accuracy Analysis for High Dimensional Weighted Distance

We now give the analysis of our Algorithm 7 for high dimensional weighted $\ell_1$-distance query.

---

**Algorithm 7** High-dimensional weighted $\ell_1$ distance query

---

1: **datastrucutre** DPTREEHIGHDIM                                          ▷ Theorem H.3
2: **members**
3:      $\mathcal{D}_1, \ldots, \mathcal{D}_d$ : DPTREEDISTANCE                          ▷ Alg. 5
4:      $X : [0, R]^{n \times d}$
5:      $w : [-R_w, R_w]^n$
6: **end members**
7: **procedure** INIT($X \subset [0, R]^d$, $n \in \mathbb{N}_+$, $w \in [-R_w, R_w]^n$, $\epsilon \in (0, 1)$, $\delta \in (0, 1)$, $\delta' \in (0, 1)$,
    $c \in (0, 0.1)$)                                                  ▷ Lemma F.2
8:      $X \leftarrow X$
9:      $w \leftarrow w$
10:     **for** $i = 1 \rightarrow d$ **do**
11:          $\mathcal{D}_i$.INIT($X_{:,i}$, $n$, $w$, $c\epsilon/\sqrt{d \log(1/\delta')}$, $\delta/d$)            ▷ Alg. 5
12:     **end for**
13: **end procedure**
14: **procedure** DISTANCEQUERY($y \in [0, R]^d$, $\alpha \in (0, 1)$)              ▷ Lemma H.1, Lemma H.2
15:     Value $\leftarrow 0$
16:     **for** $i = 1 \rightarrow d$ **do**
17:          Value $\leftarrow$ Value $+ \mathcal{D}_i$.DISTANCEQUERY($y_i, \alpha$)              ▷ Alg. 6
18:     **end for**
19:     **return** Value
20: **end procedure**
21: **end datastrucutre**

---

**Lemma H.1** (Privacy of DISTANCEQUERY, Algorithm 7). *If the following conditions hold*

- *Let data set $X \in [0, R]^{n \times d}$, weights $w \in [-R_w, R_w]^n$, query $y \in [0, R]^d$.*

- *Let $\epsilon \in (0, 1)$, $\delta \in (0, 1)$, $\delta' \in (0, 1)$.*

- *Let $c \in (0, 0.1)$ be a small constant and A be the output of DISTANCEQUERY in Algorithm 7, where each one-dimensional algorithm is configured to be $(c\epsilon/\sqrt{d \log(1/\delta')}, \delta/d)$-DP (see Line 11).*

- *Let $A_* = \sum_{i \in [n]} w_i \|y - x_i\|_1$ represent the true distance query value.*

- *Let $\epsilon = O(\log(1/\delta'))$.*

*Then, we have the output process of DISTANCEQUERY (Algorithm 7) is $(\epsilon, \delta + \delta')$-DP.*

*Proof.* The $(\epsilon, \delta + \delta')$-DP guarantee follows from the approximate DP advanced composition result Theorem C.8. Our algorithm instantiate each one-dimensional data structure with $(c\epsilon/\sqrt{d \log(1/\delta')}, \delta/d)$-DP total $d$ times.

From advanced composition in Theorem C.8, for a sufficient small parameter $\epsilon$ and constant $c$, we have the final privacy loss parameter be:

$$O(c\epsilon\sqrt{2d \log(1/\delta')}/\sqrt{d \log(1/\delta')}) = O(\epsilon)$$

and the final failure probability parameter be:

$$d\delta/d + \delta' = \delta + \delta'.$$

$\square$

**Lemma H.2** (Accuracy of DISTANCEQUERY, Algorithm 7). *If the following conditions hold*

- *Let data set $X \in [0, R]^{n \times d}$, weights $w \in [-R_w, R_w]^n$, query $y \in [0, R]^d$.*

- *Let $\epsilon \in (0, 1)$, $\delta \in (0, 1)$, $\delta' \in (0, 1)$.*

- *Let $c \in (0, 0.1)$ be a small constant and A be the output of DISTANCEQUERY in Algorithm 7, where each one-dimensional algorithm is configured to be $(c\epsilon/\sqrt{d \log(1/\delta')}, \delta/d)$-DP (see Line 11).*

- *Let $A_* = \sum_{i \in [n]} w_i \|y - x_i\|_1$ represent the true distance query value.*

*With probability $0.99$, we have*

$$|A - A_*| \leq \alpha A_* + O(\epsilon^{-1} \alpha^{-1/2} R R_w d \sqrt{\log(1/\delta')} \cdot \log^{3/2} n).$$

*Proof.* Let $A_i$ be the $i$-th dimension output returned by $\mathcal{D}_i$ in Algorithm 7. Let $A_{*,i}$ be the true distance query value in the $i$-th dimension. Observe that $A_* = \sum_{i=1}^{d} A_{*,i}$ and $A = \sum_{i=1}^{d} A_i$.

We follow the similar idea in the proof of Lemma G.5. Let $z_{j,i}$ be the random variables that represent $z_j$ (used in the proof of Lemma G.5) for the $i$-th coordinate. We can observe that the overall error across $d$ coordinates can be upper bounded by

$$|\sum_{i=1}^{d} \sum_{j=0}^{O(\log(n)/\alpha)} \frac{R z_{j,i}}{(1+\alpha)^j}|$$

where each $z_{j,i}$ is the sum of $O(\log n)$ truncated Laplace random variables independent to others. With $\epsilon$ scaled down by $c\epsilon/\sqrt{d \log(1/\delta')}$ and $\delta$ scaled down by $\delta/d$, the variance of each individual dimension is given by (see Eq. (5))

$$O(\alpha^{-1} \epsilon^{-2} d R^2 R_w^2 \log(1/\delta') \log^3 n).$$

Thus, the total variance for $d$ instantiated data structures is then

$$O(\alpha^{-1} \epsilon^{-2} d^2 R^2 R_w^2 \log(1/\delta') \log^3 n).$$

Finally, from Lemma C.3, we have the additive error given by

$$O(\alpha^{-1/2} \epsilon^{-1} d R R_w \sqrt{\log(1/\delta')} \cdot \log^{3/2} n).$$

$\square$

## H.2 High Dimension Single Data Structure

We have the data structure that can solve weighted $\ell_1$-distance problem in $d$-dimensional data.

**Theorem H.3** (DPTREEHIGHDIM data structure). *There is a data structure* DPTREEHIGHDIM *(Algorithm 7) that uses $O(nd)$ spaces to solve weighted $\ell_1$-distance query problem for dataset $X \subset [0, R]^d$ and support the following operations:*

- INIT($X \subset [0, R]^d, n \in \mathbb{N}_+, w \in [-R_w, R_w]^n, \epsilon \in (0, 1), \delta \in (0, 1), \delta' \in (0, 1), c \in (0, 0.1)$). *(Algorithm 7) It takes $O(nd)$ time to initialize the data structure.*

- DISTANCEQUERY($y \in [0, R]^d, \alpha \in (0, 1)$). *(Algorithm 7) It takes $O(\alpha^{-1} d \log^2 n)$ time to output a number $z$ such that*

  - *the process of output $z$ satisfies is $(\epsilon, \delta + \delta')$-DP private, which computes $\sum_{i \in [n]} w_i \|y - x_i\|_1$,*
  - *$|z - \sum_{i \in [n]} w_i \|y - x_i\|_1| \leq \alpha \sum_{i \in [n]} w_i \|y - x_i\|_1 + O(\epsilon^{-1} \alpha^{-1/2} R R_w d \sqrt{\log(1/\delta')} \cdot \log^{3/2} n)$,*
  - *it holds with probability $0.99$.*

*Proof.* For the runtime analysis, since we loop data structure DPTREEDISTANCE $d$ times, an additional $d$ factor will appear for both initialization and query time complexity. The DP is proved by Lemma H.1. The accuracy is proved by Lemma H.2. $\square$

# I Adaptive Query

In this section, we introduce how we can solve the adaptive query problem by our algorithm, using some tools from Qin et al. [2022]. Our idea is that, if we can prove that our algorithm can solve any query in the query space with certain error. Then, since adaptive query must lie in this space, we can handle adaptive query. In Section I.1, we show how we can boost the constant probability of our algorithm to high probability. In Section I.2, we show how we can apply the notion of $\epsilon_0$-net and bound all query points in net. In Section I.3, we show how we can bound all points in the query space by introducing an additive error. In Section I.4, we examine the effects of different norms on our adaptive query proof.

First, from Theorem H.3, given query $y \in [0, R]^d, \alpha \in (0, 1)$ we have DISTANCEQUERY$(y, \alpha)$ that can solve $d$-dimension weighted $\ell_1$-distance problem with constant probability $0.99$. Now we show how to improve it to solve adaptive query problem.

## I.1 Boost the Constant Probability to High Probability

We can repeat the data structure multiple times and take the median to boost the constant probability using Chernoff bound from Lemma C.2.

**Lemma I.1** (Using Chernoff bound to boost the probability). *If the following conditions hold:*

- *Let data set $X \in [0, R]^{n \times d}$, weights $w \in [-R_w, R_w]^n$, query $y \in [0, R]^d$.*

- *Let relative error parameter $\alpha \in (0, 1)$, the failure probability $p_f \in (0, 0.01)$.*

- *We create $l = O(\log(1/p_f))$ independent copies of data structure DPTREEHIGHDIM and take the median of the outputs with each data structure instantiated with $(\epsilon/l, (\delta + \delta')/l)$-DP.*

- *Let $A_* = \sum_{i \in [n]} w_i \|y - x_i\|_1$ be the true answer.*

- *Let $B = O(\epsilon^{-1} \alpha^{-1/2} l R R_w d \sqrt{\log(l/\delta')} \cdot \log^{3/2} n)$.*

*Then for each fixed query point $y$, we can have the process of outputting the median of $l$ responses is $(\epsilon, \delta + \delta')$-DP and the error is upper bounded by $\alpha A_* + B$ with probability $1 - p_f$.*

*Proof.* By basic composition Fact C.6, we prove the DP. Similar to the proof of Theorem E.2, we prove the error by Chernoff bound (Lemma C.2). $\square$

## I.2 From Each Fixed Query Point to All On-net Points

In this section, we build $\epsilon_0$-net and generalize from each fixed query point to all on-net points.

**Definition I.2** ($\ell_p$ $\epsilon_0$-net, see Definition 4.2.1 in Vershynin [2017]). *We define $N$ be $\ell_p$ $\epsilon_0$-net of $\mathcal{B} := \{q \in [0, R]^d\}$ such that, for every point $q$ in $\mathcal{B}$, there exists $y \in N$ satisfying $\|y - q\|_p \leq \epsilon_0$.*

**Fact I.3** ($\ell_\infty$ $\epsilon_0$-net). *Let $N$ be the $\ell_\infty$ $\epsilon_0$-net of $\mathcal{B}$, and $|N|$ be the size of net $N$. We have $|N| \leq (5R/\epsilon_0)^d$.*

**Fact I.4** ($\ell_2$ $\epsilon_0$-net, see Lemma 5 in Woodruff [2014]). *Let $N$ be the $\ell_2$ $\epsilon_0$-net of $\mathcal{B}$, and $|N|$ be the size of net $N$. We have $|N| \leq (5R/\epsilon_0)^d$.*

**Fact I.5** ($\ell_1$ $\epsilon_0$-net, see Theorem 2 in Guntuboyina and Sen [2012]). *Let $N$ be the $\ell_1$ $\epsilon_0$-net of $\mathcal{B}$, and $|N|$ be the size of net $N$. We have $|N| \leq (5R\sqrt{d}/\epsilon_0)^d$.*

**Lemma I.6** (From for each query point to for all points in net). *If the following conditions hold:*

- *Let $N$ be the $\ell_\infty$ $\epsilon_0$-net of $\mathcal{B}$, and $|N|$ be the size of net $N$.*

- *Let data set $X \in [0, R]^{n \times d}$, weights $w \in [-R_w, R_w]^n$, query $y \in [0, R]^d$.*

- *Let relative error parameter $\alpha \in (0, 1)$, the failure probability $p_f \in (0, 0.01)$.*

- *We create $l = O(\log(|N|/p_f))$ independent copies of data structure DPTREEHIGHDIM and take the median of the outputs with each data structure instantiated with $(\epsilon/l, (\delta + \delta')/l)$-DP.*

- *Let $A_* = \sum_{i \in [n]} w_i \|y - x_i\|_1$ be the true answer.*

- *Let $B = O(\epsilon^{-1}\alpha^{-1/2} l R R_w d \sqrt{\log(l/\delta')} \cdot \log^{3/2} n)$.*

*Then with probability $1 - p_f$, for all query points $y \in N$, we can have the process of outputting the median of $l$ responses is $(\epsilon, \delta + \delta')$-DP and the error is upper bounded by $\alpha A_* + B$.*

*Proof.* By basic composition Fact C.6, we prove the DP. From Lemma I.1, we know for each $y \in N$, the error is upper bounded by $\alpha A_* + B$ with probability $1 - p_f/|N|$.

Then, by union bound, with probability $1 - p_f$, the error of all $|N|$ query points in the net $y \in N$ is upper bounded by $\alpha A_* + B$. □

### I.3 From Net Points to All Points

In this section, we show how to generalize points from net to all points in the query space. Since adaptive query must lie in this space, we complete the proof of adaptive query.

**Lemma I.7** (Lipschitz of query function)**.** *If the following conditions hold:*

- *Let data set $X \in [0, R]^{n \times d}$, weights $w \in [-R_w, R_w]^n$, query $y \in [0, R]^d$.*

- *Let $Z(y) := \sum_{i \in [n]} w_i \|y - x_i\|_1$.*

- *Let $L = nR_w$.*

*Then, we have $Z(y)$ is L-Lipschitz (note that we have $\ell_1$ Lipschitz here).*

*Proof.* We can show

$$
\begin{aligned}
|Z(y) - Z(\widetilde{y})| &= |\sum_{i \in [n]} w_i \|y - x_i\|_1 - \sum_{i \in [n]} w_i \|\widetilde{y} - x_i\|_1| \\
&\leq \sum_{i \in [n]} |w_i| \cdot |\|y - x_i\|_1 - \|\widetilde{y} - x_i\|_1| \\
&\leq \sum_{i \in [n]} |w_i| \cdot \|y - \widetilde{y}\|_1 \\
&= nR_w \cdot \|y - \widetilde{y}\|_1
\end{aligned}
$$

where the first step follows from definition of $Z(y)$, the second step follows from triangular inequality, the third step follows from reverse triangular inequality, the fourth step follows from $w \in [-R_w, R_w]^n$. □

**Lemma I.8** (From points in net to all points in query space)**.** *If the following conditions hold:*

- *Let $N$ be the $\ell_\infty$ $\epsilon_0$-net of $\mathcal{B}$, and $|N|$ be the size of net $N$.*

- *Let data set $X \in [0, R]^{n \times d}$, weights $w \in [-R_w, R_w]^n$, query $y \in [0, R]^d$.*

- *Let relative error parameter $\alpha \in (0, 1)$, the failure probability $p_f \in (0, 0.01)$.*

- *We create $l = O(\log((R/\epsilon_0)^d/p_f))$ independent copies of data structure $\{\text{DPTREEHIGHDIM}_j\}_{j=1}^l$ and take the median of the outputs with each data structure instantiated with $(\epsilon/l, (\delta + \delta')/l)$-DP.*

- *Let $f(y) := \text{Median}(\{\text{DPTREEHIGHDIM}_j.\text{DISTANCEQUERY}(y, \alpha)\}_{j=1}^l)$.*

- *Let $Z(y) := \sum_{i \in [n]} w_i \|y - x_i\|_1$, where $Z(y)$ is L-Lipschitz with $L = nR_w$.*

- *Let $B = O(\epsilon^{-1}\alpha^{-1/2}lRR_w d\sqrt{\log(l/\delta')} \cdot \log^{3/2} n)$.*

*Then with probability $1 - p_f$, for all query points $q \in \mathcal{B}$, there exists a point $y \in N$ which is the closest to $q$, we can have the process of outputting the median of $l$ responses is $(\epsilon, \delta + \delta')$-DP and the error satisfy*

$$|f(y) - Z(q)| \leq \alpha Z(q) + B + 2Ld\epsilon_0.$$

*Proof.* By basic composition Fact C.6, we prove the DP.

We define an event $E$ such that:

$$\forall y \in N$$
$$|f(y) - Z(y)| \leq \alpha Z(y) + B.$$

From Lemma I.1, with $l = O(\log(|N|/p_f))$ we know

$$\Pr[\text{event } E \text{ holds}] \geq 1 - p_f$$

We can show

$$l = O(\log(|N|/p_f)$$
$$= O(\log((R/\epsilon_0)^d/p_f)$$

where the first step follows from definition of $l$, the second step follows from Fact I.3.

We condition on event $E$ to be held. Then, by definition of $\ell_\infty$ $\epsilon_0$-net (see Definition I.2), for each $q \notin N$, there exists $y \in N$ such that

$$\|y - q\|_\infty \leq \epsilon_0 \tag{6}$$

We know

$$|Z(y) - Z(q)| \leq L \cdot \|y - q\|_1$$
$$\leq L \cdot d\|y - q\|_\infty$$
$$\leq L \cdot d\epsilon_0 \tag{7}$$

where the first step follows from Lemma I.7, the second step follows from $\|x\|_1 \leq d\|x\|_\infty$ for $x \in \mathbb{R}^d$, and the last step follows from Eq. (6).

Using the on-net query $y$ to answer the off-net query $q$, for any $q \notin N$, we have

$$|f(y) - Z(q)| \leq |f(y) - Z(y)| + |Z(q) - Z(y)|$$
$$\leq |f(y) - Z(y)| + L \cdot d \cdot \epsilon_0$$
$$\leq \alpha Z(y) + B + L \cdot d \cdot \epsilon_0$$
$$\leq \alpha Z(q) + B + 2L \cdot d \cdot \epsilon_0 \tag{8}$$

where the first step follows from triangular inequality, the second step follows from Eq. (7), the third step follows from Lemma I.6, and the last step follows from Eq. (7).

Thus, we complete the proof. $\square$

Therefore, even adaptive queries can be answered accurately, since any adaptive query can be assumed in $\mathcal{B}$.

### I.4 Effect of Different Norms on the Result

In the above proof, we have two different measure spaces, i.e. $\ell_\infty$ distance of $\epsilon_0$-net (Definition I.2) and $\ell_1$ Lipschitz (Lemma I.7).

One might ask, will the norm we choose in two spaces have an impact on the final result? We can show that the norm we choose currently is sufficient to use.

For different norms, the only differences in the proofs will be Lipschitz smoothness in Eq. (7) and the cardinality of $\epsilon_0$-net, i.e. $|N|$ in Fact I.3.

**Lemma I.9.** *If we use $\ell_\infty$ $\epsilon_0$-net and use $\ell_1$ Lipschitz in Lemma I.8, we have copies of data structure $l = O(d \log(nR/p_f))$.*

*Proof.* If we use $\ell_\infty$ to bound the distance to net, Eq. (7) is:

$$
\begin{aligned}
|Z(y) - Z(q)| &\leq nR_w \cdot \|y - q\|_1 \\
&\leq nR_w \cdot d\|y - q\|_\infty \\
&\leq nR_w \cdot d\epsilon_0
\end{aligned}
$$

where the first step follows from Lemma I.7, the second step follows from $\|x\|_1 \leq d\|x\|_\infty$ for $x \in \mathbb{R}^d$, and the last step follows from $\ell_\infty$ $\epsilon_0$-net.

Then, Eq. (8) is

$$
|f(y) - Z(q)| \leq \alpha Z(q) + B + 2nR_w \cdot d \cdot \epsilon_0
$$

For $\ell_\infty$ distance, we have $|N| \leq (5R/\epsilon_0)^d$ in Fact I.3.

We can choose $\epsilon_0 = \Theta(1/n)$ to hide $nR_w \cdot d \cdot \epsilon_0$ term in $B$ in Lemma I.8. Thus,

$$
\begin{aligned}
l &= O(\log(|N|/p_f) \\
&= O(\log((R/\epsilon_0)^d/p_f) \\
&= O(\log((nR)^d/p_f)) \\
&= O(d \log(nR/p_f))
\end{aligned}
$$

where the last step follows from $\log(a^d/b) = O(d \log(a/b))$ for any $a > 1, 0 < b < 1, d > 1$. $\quad\square$

**Lemma I.10.** *If we use $\ell_2$ $\epsilon_0$-net and use $\ell_1$ Lipschitz in Lemma I.8, we have copies of data structure $l = O(d \log(nR/p_f))$.*

*Proof.* If we use $\ell_2$ to bound the distance to net, Eq. (7) changes to be:

$$
\begin{aligned}
|Z(y) - Z(q)| &\leq nR_w \cdot \|y - q\|_1 \\
&\leq nR_w \cdot \sqrt{d} \cdot \|y - q\|_2 \\
&\leq nR_w \cdot \epsilon_0 \sqrt{d}
\end{aligned}
$$

where the first step follows from Lemma I.7, the second step follows from $\|x\|_1 \leq \sqrt{d} \cdot \|x\|_2$ for $x \in \mathbb{R}^d$, and the last step follows from $\ell_2$ $\epsilon_0$-net.

Then, Eq. (8) changes to be

$$
|f(y) - Z(q)| \leq \alpha Z(q) + B + 2nR_w \cdot \epsilon_0 \sqrt{d}
$$

For $\ell_2$ distance, we also have $|N| \leq (5R/\epsilon_0)^d$ in Fact I.4.

We can choose $\epsilon_0 = \Theta(1/n)$ to hide $nR_w \cdot \sqrt{d} \cdot \epsilon_0$ term in $B$ in Lemma I.8. Thus,

$$
\begin{aligned}
l &= O(\log(|N|/p_f) \\
&= O(\log((R/\epsilon_0)^d/p_f) \\
&= O(\log((nR)^d/p_f)) \\
&= O(d \log(nR/p_f))
\end{aligned}
$$

where the last step follows from $\log(a^d/b) = O(d \log(a/b))$ for any $a > 1, 0 < b < 1, d > 1$. $\quad\square$

**Lemma I.11.** *If we use $\ell_1$ $\epsilon_0$-net and use $\ell_1$ Lipschitz in Lemma I.8, we have copies of data structure $l = O(d \log(ndR/p_f))$.*

*Proof.* If we use $\ell_1$ to bound the distance to net, Eq. (7) changes to be:

$$|Z(y) - Z(q)| \leq nR_w \cdot \|y - q\|_1$$
$$\leq nR_w \cdot \epsilon_0$$

where the first step follows from Lemma I.7, and the last step follows from $\ell_1$ $\epsilon_0$-net.

Then, Eq. (8) changes to be

$$|f(y) - Z(q)| \leq \alpha Z(q) + B + 2nR_w \cdot \epsilon_0$$

For $\ell_1$ distance, we have $|N| \leq (5R\sqrt{d}/\epsilon_0)^d$.

We can choose $\epsilon_0 = \Theta(1/n)$ to hide $nR_w \cdot \epsilon_0$ term in $B$ in Lemma I.8. Thus,

$$l = O(\log(|N|/p_f)$$
$$= O(\log((R\sqrt{d}/\epsilon_0)^d/p_f)$$
$$= O(\log((nR\sqrt{d})^d/p_f))$$
$$= O(d\log(nRd/p_f))$$

where the last step follows from $\log(a^d/b) = O(d\log(a/b))$ for any $a > 1, 0 < b < 1, d > 1$. $\qquad\square$

From the above analysis, we can show that $\ell_\infty$ or $\ell_2$ $\epsilon_0$-net is slightly better than $\ell_1$ $\epsilon_0$-net.

- $\ell_\infty$ $\epsilon_0$-net, Lemma I.9: we have $l = O(d\log(nR/p_f))$.
- $\ell_2$ $\epsilon_0$-net, Lemma I.10: we have $l = O(d\log(nR/p_f))$.
- $\ell_1$ $\epsilon_0$-net, Lemma I.11: we have $l = O(d\log(nRd/p_f))$.

Thus, the norm we choose for $\epsilon_0$-net is sufficient good.

## J  Softmax Activation

In this section, we introduce how we extend previous $\ell_1$ distance results to the Softmax activation function, which is the most widely used distance measure in attention mechanism based models.

In Section J.1, we show how to extend to the Softmax distance function in Lemma J.6. In Section J.2, we show how to adjust our algorithms. In Section J.3, we extend our algorithm to be robust to adaptive query. In Section J.4, we give the proof of our main result Theorem 3.1.

### J.1  Exponential Inner Product

In this section, we show how we obtain the Softmax distance using $\ell_2^2$ distance query. First, we provide some helpful results from Alman and Song [2023].

**Definition J.1** (Definition 3.1 in Alman and Song [2023]). *Let $r \geq 1$ denote a positive integer. Let $\epsilon \in (0, 0.1)$ denote an accuracy parameter. Given a matrix $A \in \mathbb{R}_{\geq 0}^{n \times n}$, we say $\widetilde{A} \in \mathbb{R}_{\geq 0}^{n \times n}$ is an $(\epsilon, r)$-approximation of $A$ if*

- *$\widetilde{A} = U_1 \cdot U_2^\top$ for some matrices $U_1, U_2 \in \mathbb{R}^{n \times r}$ (i.e., $\widetilde{A}$ has rank at most $r$), and*

- *$|\widetilde{A}_{i,j} - A_{i,j}| \leq \epsilon \cdot A_{i,j}$ for all $(i, j) \in [n]^2$.*

**Lemma J.2** (Lemma 3.4 in Alman and Song [2023]). *Suppose $Q, K \in \mathbb{R}^{n \times d}$, with $\|Q\|_\infty \leq R$, and $\|K\|_\infty \leq R$. Let $A := \exp(QK^\top/d) \in \mathbb{R}^{n \times n}$. For accuracy parameter $\epsilon \in (0, 0.1)$, there is a positive integer $s$ bounded above by*

$$s = O\Big(\max\Big\{\frac{\log(1/\epsilon)}{\log(\log(1/\epsilon)/R)}, R^2\Big\}\Big), \tag{9}$$

*and a positive integer $r$ bounded above by*

$$r \leq \binom{2s + 2d}{2s} \tag{10}$$

*such that: There is a matrix $\widetilde{A} \in \mathbb{R}^{n \times n}$ that is an $(\epsilon, r)$-approximation (Definition J.1) of $A \in \mathbb{R}^{n \times n}$. Furthermore, the matrices $U_1$ and $U_2$ defining $\widetilde{A}$ can be computed in $O(n \cdot r)$ time.*

Here we consider the vector version of Lemma J.2.

**Definition J.3.** *We define $\Gamma_{R,s} := \max_{j \in [s]} \frac{R^j}{\sqrt{j!}}$.*

Then, we have $P(x) : [0, R]^d \rightarrow [0, \Gamma_{R,s}]^r$ where $P(\cdot)$ is polynomial kernel function defined in Alman and Song [2023].

**Remark J.4.** *We use $\Gamma_{R,s}$ to denote the value range of our polynomial kernel methods function, i.e., $P(x) : [0, R]^d \rightarrow [0, \Gamma_{R,s}]^r$. The factorial term in $\Gamma_{R,s}$ comes from Taylor approximation coefficients. We take the maximum overall $s$ order approximation terms to get the upper bound of our value range.*

**Lemma J.5** (Polynomial approximation). *For any accuracy parameter $\epsilon_s \in (0, 0.1)$, let $R \geq 1$, and let $P(x) : [0, R]^d \rightarrow [0, \Gamma_{R,s}]^r$ be the $s$-th order polynomial kernel function defined in Alman and Song [2023] where $r \leq \binom{2s+2d}{2s}$ and $s = O(\max\{\frac{\log(1/\epsilon_s)}{\log(\log(1/\epsilon_s)/R)}, R^2\})$. Then, for any $x, y \in [0, R]^d$, we have*

$$|P(x)^\top P(y) - \exp(x^\top y/d)| \leq \epsilon_s \cdot \min\{\exp(x^\top y/d), P(x)^\top P(y)\}$$

*Furthermore, the vectors $P(x)$ and $P(y)$ can be computed in $O(r)$ time.*

*Proof.* Let $n = 1$. The proof follows from directly applying Lemma J.2. $\qquad\square$

Using the results from Alman and Song [2023] above, we can extend our results to Softmax activation.

**Lemma J.6** (Weighted Softmax approximation, formal version of Lemma D.7). *Let accuracy parameter be $\epsilon_s \in (0, 0.1)$. Let $R \geq 1$. Let $r \leq \binom{2s+2d}{2s}$ and $s = O(\max\{\frac{\log(1/\epsilon_s)}{\log(\log(1/\epsilon_s)/R)}, R^2\})$. Let $P(x) : [0, R]^d \rightarrow [0, \Gamma_{R,s}]^r$ be the $s$-th order polynomial kernel function defined in Lemma J.5. Then we can approximate exponential inner product using polynomial kernel function:*

$$w^\top \exp(Xy/d) = -\frac{1}{2} \sum_{j \in [r]} \sum_{i \in [n]} w_i |P(x_i)_j - P(y)_j|^2 + \frac{1}{2} \sum_{i \in [n]} w_i (\|P(x_i)\|_2^2 + \|P(y)\|_2^2)$$
$$+ O(w^\top \exp(Xy/d) \cdot \epsilon_s)$$

*Moreover, the vectors $P(\cdot)$ can be computed in $O(r)$ time.*

*Proof.* From Lemma J.5, we can use polynomial kernel to approximate the Softmax function:

$$w^\top \exp(Xy/d) = \sum_{i \in [n]} w_i P(x_i)^\top P(y) + O(w^\top \exp(Xy/d) \cdot \epsilon_s).$$

The proof of approximation error and time complexity of constructing $P(\cdot)$ follows from Lemma J.5.

Then, we can show

$$2 \sum_{i \in [n]} w_i P(x_i)^\top P(y) = -\sum_{i \in [n]} w_i \|P(x_i) - P(y)\|_2^2 + \sum_{i \in [n]} w_i (\|P(x_i)\|_2^2 + \|P(y)\|_2^2)$$
$$= -\sum_{j \in [r]} \sum_{i \in [n]} w_i |P(x_i)_j - P(y)_j|^2 + \sum_{i \in [n]} w_i (\|P(x_i)\|_2^2 + \|P(y)\|_2^2)$$

where the first step follows from $\|x - y\|_2^2 = \|x\|_2^2 + \|y\|_2^2 - 2\langle x, y \rangle$, and the second step follows $\|x\|_2^2 = \sum_{j=1}^d |x_j|^2$ for $x \in \mathbb{R}^d$. $\qquad\square$

## J.2 Algorithm Modifications

Based on Lemma J.6, we can now extend our DP algorithms to handle Softmax activation. First, we need to construct $P(y)$ and $P(x_i)$ for $i \in [n]$, each costing $O(r)$ time. Then, for the second term in Lemma J.6, i.e. $\frac{1}{2} \sum_{i \in [n]} w_i(\|P(x_i)\|_2^2 + \|P(y)\|_2^2)$, we don't need to add DP noises in it; instead, we calculate this term exactly, preprocess it, and store the results in the algorithm. For the first term, $-\frac{1}{2} \sum_{j \in [r]} \sum_{i \in [n]} w_i |P(x_i)_j - P(y)_j|^2$, we can adjust our high dimensional DP distance query algorithm to solve it.

Due to the decomposability of $\ell_p^p$ norm, i.e.

$$\sum_{i \in [n]} w_i \|x_i - y\|_p^p = \sum_{j \in [d]} \sum_{i \in [n]} w_i |x_{i,j} - y_j|^p,$$

we can compute $\ell_2^2$ norm easily (see details in Lemma F.2). We then show how to extend our one dimensional $\ell_1$ distance algorithm (Algorithm 5 and 6) to $\ell_2^2$ distance with minor modifications.

**Theorem J.7** (DPTREEDISTANCE $\ell_2^2$ distance). *With $\alpha$ scaled down by a factor of $2$ and all QUERY instead multiplied by $R^2/(1 + \alpha/2)^{2j}$ in Lines 8 and 13 of Algorithm 6, i.e., from*

- *Lines 8 and 13: Value $\leftarrow$ Value + $\mathcal{D}$.QUERY$(l_j, r_j) \cdot \frac{R}{(1+\alpha)^j}$*

*to*

- *Lines 8 and 13: Value $\leftarrow$ Value + $\mathcal{D}$.QUERY$(l_j, r_j) \cdot \frac{R^2}{(1+\alpha/2)^{2j}}$.*

*The data structure DPTREEDISTANCE (Algorithm 5,6) uses $O(n)$ spaces to solve weighted $\ell_2^2$-distance query problem for dataset $X \subset [0, R]$ and support the following operations:*

- INIT$(X \subset [0, R], n \in \mathbb{N}_+, w \in [-R_w, R_w]^n, \epsilon \in (0, 1), \delta \in (0, 1))$. *(Algorithm 5) It takes $O(n)$ time to initialize the data structure.*

- DISTANCEQUERY$(y \in [0, R], \alpha \in (0, 1))$. *(Algorithm 6)*

  *It takes $O(\alpha^{-1} \log^2 n)$ time to output a number $z$ such that*

  - *the process of output $z$ satisfies $(\epsilon, \delta)$-DP private, which computes $\sum_{i \in [n]} w_i |y - x_i|^2$,*
  - *$|z - \sum_{i \in [n]} w_i |y - x_i|^2| \leq \alpha \sum_{i \in [n]} w_i |y - x_i|^2 + O(\epsilon^{-1} \alpha^{-1/2} R^2 R_w \log^{3/2} n)$,*
  - *it holds with probability $0.99$.*

*Proof.* The proof is similar to that of Theorem G.6, except that now our additive error includes $R$ increased by a power of 2, i.e., from $O(\epsilon^{-1} \alpha^{-1/2} R R_w \log^{3/2} n)$ to $O(\epsilon^{-1} \alpha^{-1/2} R^2 R_w \log^{3/2} n)$. $\square$

Now we can give our result that can answer Softmax query.

**Theorem J.8** (Softmax query, formal version of Theorem 4.2). *Let $R \geq 1$. Let $r \leq \binom{2s+2d}{2s}$ and $s = O(\max\{\frac{\log(1/\epsilon_s)}{\log(\log(1/\epsilon_s)/R)}, R^2\})$. Let $\Gamma_{R,s}$ be defined in Definition J.3. Let accuracy parameter be $\epsilon_s \in (0, 0.1)$. There is a data structure DPTREESOFTMAX (Algorithm 2) that uses $O(nr)$ spaces to solve Softmax query problem for dataset $X \subset [0, R]^d$ and support the following operations:*

- INIT$(X \subset [0, R]^d, n \in \mathbb{N}_+, w \in [-R_w, R_w]^n, \epsilon \in (0, 1), \delta \in (0, 1), \delta' \in (0, 1), c \in (0, 0.1), \epsilon_s \in (0, 0.1))$. *(Algorithm 2) It takes $O(nr)$ time to initialize the data structure.*

- DISTANCEQUERY$(y \in [0, R]^d, \alpha \in (0, 1))$. *(Algorithm 2) It takes $O(\alpha^{-1} r \log^2 n)$ time to output a number $z$ such that*

  - *the process of output $z$ satisfies $(\epsilon, \delta + \delta')$-DP private, which computes $w^\top \exp(Xy/d)$,*
  - *$|z - w^\top \exp(Xy/d)| \leq (\alpha + \epsilon_s) \cdot w^\top \exp(Xy/d)$*
    *$+ O(\epsilon^{-1} \alpha^{-1/2} \Gamma_{R,s}^2 R_w r \sqrt{\log(1/\delta')} \cdot \log^{3/2} n)$,*

– *it holds with probability* $0.99$.

*Proof.* The DP proof is the same as the proof of Lemma H.1.

We then show the time complexity. From Lemma J.6, we know that constructing $P(\cdot)$ requires $O(r)$ time. In the first for loop of INIT, the dominating time consumption is $O(nr)$. The second for loop also has a time complexity of $O(nr)$. Therefore, the total time complexity for INIT is $O(nr)$. In the DISTANCEQUERY function, constructing $P(y)$ takes $O(r)$ time. Within the for loop, it requires $O(\alpha^{-1} r \log^2 n)$. Thus, the total time complexity for DISTANCEQUERY is $O(\alpha^{-1} r \log^2 n)$.

The space complexity is $O(nr)$, since storing the $n \times r$ matrix $P$ is the dominating factor.

The proof of the error follows from the triangle inequality by combining the errors in Lemma J.6 and Theorem J.7. $\square$

## J.3 Adaptive Softmax

In this section, we show how to make Algorithm 2 robust to adaptive query. We follow the same idea from Section I. We notice that, in the Softmax activation, we have query function $Z(y) := w^\top \exp(Xy/d)$ different from the $\ell_1$-distance in Section I. Therefore, we need to re-calculate Lipschitz constant first.

**Lemma J.9** (Lipschitz of weighted Softmax)**.** *If the following conditions hold:*

- *Let data set $X \in [0, R]^{n \times d}$, weights $w \in [-R_w, R_w]^n$, query $y \in [0, R]^d$.*

- *Let $Z(y) := w^\top \exp(Xy/d)$.*

- *Let $L = nd^{-1/2} R R_w \exp(R^2)$.*

*Then, we have $Z(y)$ is $L$-Lipschitz (note that we have $\ell_1$ Lipschitz here).*

*Proof.* We can show

$$
\begin{aligned}
|Z(y) - Z(\widetilde{y})| &= |\sum_{i \in [n]} w_i \exp(x_i^\top y/d) - \sum_{i \in [n]} w_i \exp(x_i^\top \widetilde{y}/d)| \\
&\leq \sum_{i \in [n]} |w_i| \cdot |\exp(x_i^\top y/d) - \exp(x_i^\top \widetilde{y}/d)| \\
&\leq \sum_{i \in [n]} |w_i| \exp(R^2) |x_i^\top y/d - x_i^\top \widetilde{y}/d| \\
&\leq \sum_{i \in [n]} |w_i| \exp(R^2) \|x_i\|_2 \cdot \|y - \widetilde{y}\|_2/d \\
&\leq nR_w \exp(R^2) \sqrt{d} R \cdot \|y - \widetilde{y}\|_2/d \\
&\leq nd^{-1/2} R R_w \exp(R^2) \|y - \widetilde{y}\|_1
\end{aligned}
$$

where the first step follows from definition of $Z(y), Z(\widetilde{y})$, the second step follows from triangular inequality, the third step follows from Fact C.4, the fourth step follows from Cauchy–Schwarz inequality $|u^\top v| \leq \|u\|_2 \cdot \|v\|_2$ for $u, v \in \mathbb{R}^d$, the fifth step follows from $w_i \in [-R_w, R_w]$ and $x_i \in [0, R]^d$, and the last step follows from $\|u\|_2 \leq \|u\|_1$ for $u \in \mathbb{R}^d$. $\square$

Then we can show how to extend our algorithm to be robust to adaptive query.

**Lemma J.10** (Adaptive Softmax, formal version of Lemma D.8)**.** *If the following conditions hold:*

- *Let $N$ be the $\ell_\infty$ $\epsilon_0$-net of $\mathcal{B}$, and $|N|$ be the size of net $N$.*

- *Let data set $X \in [0, R]^{n \times d}$, weights $w \in [-R_w, R_w]^n$, query $y \in [0, R]^d$.*

- *Let relative error parameter $\alpha \in (0, 1)$, the failure probability $p_f \in (0, 0.01)$.*

- *We create $l = O(\log((R/\epsilon_0)^r/p_f))$ independent copies of data structure $\{\text{DPTREESOFTMAX}_j\}_{j=1}^l$ (Algorithm 2) and take the median of the outputs with each data structure instantiated with $(\epsilon/l, (\delta + \delta')/l)$-DP.*

- *Let $f(y) := \text{Median}(\{\text{DPTREESOFTMAX}_j.\text{DISTANCEQUERY}(y, \alpha)\}_{j=1}^l)$.*

- *Let $Z(y) := w^\top \exp(Xy/d)$, where $Z(y)$ is L-Lipschitz with $L = nd^{-1/2}RR_w\exp(R^2)$.*

- *Let $B = O(\epsilon^{-1}\alpha^{-1/2}l\Gamma_{R,s}^2 R_w r\sqrt{\log(l/\delta')} \cdot \log^{3/2} n)$.*

*Then with probability $1 - p_f$, for all query points $q \in \mathcal{B}$, there exists a point $y \in N$ which is the closest to q, we can have the process of outputting the median of l responses is $(\epsilon, \delta + \delta')$-DP and the error satisfies*

$$|f(y) - Z(q)| \le (\alpha + \epsilon_s)Z(q) + B + 2n\sqrt{d}RR_w\exp(R^2)\epsilon_0.$$

*Proof.* The proof follows from the same idea as the proof of Lemma I.8, except that we use Theorem J.8 and the Lipschitz in Lemma J.9. □

**Theorem J.11** (Adaptive query Softmax data structure, formal version of Theorem 4.4). *Let $R \ge 1$. Let $r \le \binom{2s+2d}{2s}$ and $s = O(\max\{\frac{\log(1/\epsilon_s)}{\log(\log(1/\epsilon_s)/R)}, R^2\})$. Let $\Gamma_{R,s}$ be defined in Definition J.3. Let accuracy parameter be $\epsilon_s \in (0, 0.1)$. Let $X \in [0, R]^{n \times d}$ be the dataset, $w \in [-R_w, R_w]^n$ be weights, $y \in [0, R]^d$ be the query, $\alpha \in (0, 1)$ be the relative error parameter, and $p_f$ be the failure probability parameter. Let $l = O(r\log(dR/(\epsilon_s p_f)))$. There is a data structure DPTREESOFT-MAXADAPTIVE (Algorithm 1) that uses $O(lnr)$ spaces to solve weighted Softmax query problem for dataset $X \subset [0, R]^d$ and support the following operations:*

- INIT($X \subset [0, R]^d, n \in \mathbb{N}_+, w \in [-R_w, R_w]^n, \epsilon \in (0,1), \delta \in (0,1), \delta' \in (0,1), c \in (0, 0.1), \epsilon_s \in (0, 0.1), p_f \in (0, 0.01)$). *(Algorithm 1) It takes $O(lnd)$ time to initialize the data structure.*

- DISTANCEQUERY($y \in [0, R]^d, \alpha \in (0, 1)$). *(Algorithm 1) It takes $O(\alpha^{-1}ld\log^2 n)$ time to output a number z such that*

  - *the process of output z satisfies $(\epsilon, \delta + \delta')$-DP private, which computes $w^\top \exp(Xy/d)$,*
  - *$|z - w^\top \exp(Xy/d)| \le (\alpha + \epsilon_s) \cdot w^\top \exp(Xy/d)$ $+ O(\epsilon^{-1}\alpha^{-1/2}l\Gamma_{R,s}^2 R_w r\sqrt{\log(l/\delta')} \cdot \log^{3/2} n)$,*
  - *it holds with probability $1 - p_f$ (where $p_f$ is used in l),*
  - *it is robust to adaptive query.*

*Proof.* We only need to show how to pick $\epsilon_0$ in the parameter $l$, because everything else is the same as Lemma J.10. We know the additive error introduced by adaptive query is $E_a := O(n\sqrt{d}RR_w\exp(R^2)\epsilon_0)$ and the relative error introduced by polynomial kernel approximation is $E_p := w^\top \exp(Xy/d) \cdot \epsilon_s$. It can be shown that:

$$E_p := w^\top \exp(Xy/d) \cdot \epsilon_s$$
$$\le \epsilon_s \|w\|_2 \cdot \|\exp(Xy/d)\|_2$$
$$= O(nR_w\epsilon_s\exp(R^2))$$

where the first step follows from definition of $E_p$, the second step follows from Cauchy–Schwarz inequality, and the last step follows from $w \in [-R_w, R_w]^n$, $X \in [0, R]^{n \times d}$, and $y \in [0, R]^d$.

Picking $\epsilon_0 = \Theta(\frac{\epsilon_s}{\sqrt{d}R})$, we can hide the error of adaptive query $E_a$ in $E_p$. Thus, we have

$$l = O(\log((R/\epsilon_0)^r/p_f))$$
$$= O(\log((\sqrt{d}R^2/\epsilon_s)^r/p_f))$$
$$= O(r\log(dR/(\epsilon_s p_f)))$$

where the first step comes from the definition of $l$, the second step comes from picking $\epsilon_0 = \Theta(\frac{\epsilon_s}{\sqrt{d}R})$, and the last step follows from $\log(a^d/b) = O(d\log(a/b))$ for any $a > 1, 0 < b < 1, d > 1$. $\qquad\square$

## J.4 Proof of Main Result

In this section, we give the proof of our main result of Theorem 3.1.

**Theorem J.12** (Softmax cross-attention, formal version of Theorem 3.1). *Let $Q, K, V, \mathrm{Attn}$ be defined in Definition 1.1. Let $\alpha \in (0, 1)$ be the relative error parameter and $p_f$ be the probability of failure parameter. Let $r, s, \epsilon_s$ be parameters of polynomial kernel methods (Lemma D.7). Let $\Gamma_{R,s} := \max_{j\in[s]} \frac{R^j}{\sqrt{j!}}$ (Definition J.3). Let $l = O(r\log(dR/(\epsilon_s p_f)))$. There is a data structure* DPTREESOFTMAXADAPTIVE *(Algorithm 1) that uses $O(lnrd)$ spaces to make cross-attention DP and supports the following operations:*

- *We initialize $d$ data structures using* INIT$(K, n, V_{*,k}, \epsilon \in (0, 1), \delta \in (0, 1), \delta' \in (0, 1), c \in (0, 0.1), \epsilon_s \in (0, 0.1), p_f \in (0, 0.01))$ *(Algorithm 1), for $k \in [d]$. It takes $O(lnr)$ time to initialize one data structure.*

- *At query time, for user input $Q$, we process one token at a time by passing the $i$-th row of $Q$, denoted $Q_i \in \mathbb{R}^d$, to* DISTANCEQUERY$(Q_i, \alpha \in (0, 1))$ *(Algorithm 1) for each $i \in [m]$. It takes $O(\alpha^{-1} lr \log^2 n)$ time to output an entry $z$ in $\mathrm{Attn}(Q, K, V)$ such that*

  - *the process of output $z$ satisfies $(\epsilon, \delta + \delta')$-DP,*
  - *the process of output $z$ has relative error $n^{-1}(\alpha + \epsilon_s)$,*
  - *the process of output $z$ has additive error $O(n^{-1}\epsilon^{-1}\alpha^{-1/2}l\Gamma_{R,s}^2 R_w r\sqrt{\log(l/\delta')} \cdot \log^{3/2} n)$,*
  - *it holds with probability $1 - p_f$ (where $p_f$ is used in $l$),*
  - *it is robust to adaptive query.*

*Proof.* From Section 3, we know that we can ensure matrix $AV$ in cross-attention computation satisfies DP. Next, from Theorem 4.4, for $i \in [m], j \in [n], k \in [d]$, we have $(AV)_{i,k}$ is $(\epsilon, \delta + \delta')$-DP and also robust to adaptive query. For $D$, we compute the exact true value. By the post-processing property of DP (Fact C.5), our cross-attention process is DP.

Let $(AV)_{i,k}$ be the true value and $\widetilde{(AV)}_{i,k}$ be the noisy value. Then, from Theorem 4.4, we have

$$|(AV)_{i,k} - \widetilde{(AV)}_{i,k}| \le (\alpha + \epsilon_s) \cdot (AV)_{i,k} + O(\epsilon^{-1}\alpha^{-1/2}l\Gamma_{R,s}^2 R_w r\sqrt{\log(l/\delta')} \cdot \log^{3/2} n). \tag{11}$$

For $D_{i,i}$, we can show

$$D_{i,i} = (A \cdot \mathbf{1}_n)_i = \sum_{j=1}^{n} \exp(\langle Q_i, K_j \rangle/d) \ge n \tag{12}$$

because $\langle Q_i, K_j \rangle \ge 0$ for bounded $Q, K$.

Finally, we can show the error of one entry is bounded by

$$\begin{aligned}
|(D^{-1}AV)_{i,k} - (D^{-1}\widetilde{AV})_{i,k}| &= |D_{i,i}^{-1}((AV)_{i,k} - \widetilde{(AV)}_{i,k})| \\
&= |D_{i,i}^{-1}| \cdot |((AV)_{i,k} - \widetilde{(AV)}_{i,k})| \\
&\le n^{-1}(\alpha + \epsilon_s) \cdot (AV)_{i,k} \\
&\quad + O(n^{-1}\epsilon^{-1}\alpha^{-1/2}l\Gamma_{R,s}^2 R_w r\sqrt{\log(l/\delta')} \cdot \log^{3/2}(n))
\end{aligned}$$

where the first step follows from definition, the second step follows from simple algebra, and the last step follows from Eq.(11) and (12). $\qquad\square$

