# OpenReview forum: "Differential Privacy of Cross-Attention with Provable Guarantee"
_NeurIPS.cc/2024/Workshop/SafeGenAi — SafeGenAi Poster_

### Official Review · Reviewer_fxuK · 2024-10-09
**Good work but seems to be in different formats**

**Rating:** 7
**Confidence:** 4

**Review:**

This paper conducts a differential privacy guarantee on the LGMs theoretically and provides a detailed analysis in the appendix. The motivation is clear with enough novelty contributions for the workshop paper. However, two main concerns are listed:
1. It seems to have only theoretical results, and it suggests further providing experimental results, which can be further discussed.
2. The paper format appears to be a submission to ICRL, which should be inappropriate.

---

### Official Review · Reviewer_Wac6 · 2024-10-09
**Reviewer Wac6**

**Rating:** 6
**Confidence:** 5

**Review:**

### Summary

This paper introduces a new differential privacy data structure designed to protect the privacy of cross-attention modules in large generative models (LGMs). The authors meticulously devise a theoretical guarantee method for privacy protection in cross-attention. By defining the computational method of cross-attention and a main theorem, the paper describes how to maintain (ε, δ)-differential privacy while reducing query time and memory usage. This is the first attempt to provide differential privacy for cross-attention, potentially inspiring future algorithm designs for privacy.

### Strengths

1. **Innovativeness**: This is the first attempt to apply differential privacy to cross-attention modules, which are critical in LGMs, especially when dealing with sensitive information.
2. **Theoretical Depth**: The authors provide rigorous theoretical analysis and prove the privacy guarantees of the proposed data structure under specific conditions.
3. **Practical Application Value**: As LGMs become increasingly common in applications such as system prompts and stable diffusion, privacy protection becomes crucial. This research has significant practical application potential.
4. **Robustness**: The study also considers adaptive queries, i.e., scenarios where users may intentionally attack the cross-attention system, enhancing the practicality and robustness of the model.

### Weaknesses

1. **Lack of Experimental Validation**: The paper focuses primarily on theoretical analysis and mathematical models, lacking experimental results on real data to support the theory's effectiveness and practicality.
2. **Complexity and Practicality Issues**: Although theoretically sound, the proposed method may have high time and space complexities, potentially creating performance bottlenecks when deployed practically.
3. **Insufficient Consideration of Scalability and Universality**: The discussion lacks whether the method is applicable to different types of LGMs or performs effectively under various scenarios.

### Suggestions for Improvement

1. **Add Experimental Section**: It is recommended that the authors conduct experiments on real datasets to demonstrate the effectiveness of the proposed method in practical applications, including privacy protection effects and processing efficiency.
2. **Add Analysis of Practical Scenarios**: Discuss the application of the proposed anonymization method in practice. For example, context the latest reference [1], analyze related advantages and applicable scenarios.
3. **Performance Optimization**: Explore more potential optimization algorithms to reduce the time and space complexities of the algorithm, making it more suitable for practical application.

[1] Tianhao Huang et al., Private Language Models via Truncated Laplacian Mechanism In Proc. EMNLP 2024

---

### Official Review · Reviewer_mEf8 · 2024-10-09
**DP for Cross-Attention**

**Rating:** 6
**Confidence:** 4

**Review:**

This paper introduces a rigorous approach to ensuring differential privacy in cross-attention mechanisms. The authors leverage polynomial kernel methods and novel data structures like DP-TreeSoftmaxAdaptive to achieve (ϵ, δ)-differential privacy with bounded error guarantees, providing robustness against adaptive queries. The theoretical analysis is sound.

---

### Official Review · Reviewer_ywbc · 2024-10-11
**I think paper presents an innovative application of differential privacy to cross-attention mechanisms in large generative models, offering strong theoretical guarantees but facing challenges in empirical validation and computational scalability.**

**Rating:** 6
**Confidence:** 3

**Review:**

Pros:
1. This is innovating to apply differential privacy (DP) to cross-attention mechanisms in large generative models (LGMs), which is a novel and timely solution to the privacy concerns in AI​.

2. The paper provides rigorous mathematical proofs that ensure privacy and accuracy guarantees, including robustness to adaptive query attacks, offering a solid foundation for its proposed method​

3. The work tackles a critical issue—protecting sensitive information in AI systems using retrieval-augmented generation (RAG) and system prompts, which is highly relevant in the current AI landscape​

Cons:
1. The paper lacks extensive real-world experiments or performance comparisons with existing privacy-preserving methods, limiting insight into how well the proposed solution performs in practice​
2/. The memory and time complexity of the proposed solution may limit its scalability and efficiency in practical, large-scale AI applications​3. The method is specifically designed for cross-attention and retrieval-augmented generation (RAG) systems, which might limit its applicability to other AI models or attention mechanisms​